# FLASH MULTI-HEAD FEED-FORWARD NETWORK

## ABSTRACT

We explore Multi-Head FFN (MH-FFN) as a replacement of FFN in the Transformer architecture, motivated by the structural similarity between single-head attention and FFN. While multi-head mechanisms enhance expressivity in attention, naively applying them to FFNs faces two challenges: memory consumption scaling with the head count, and an imbalanced ratio between the growing intermediate size and the fixed head dimension as models scale, which degrades scalability and expressive power. To address these challenges, we propose Flash Multi-Head FFN (FlashMHF), with two key innovations: an I/O-aware fused kernel computing outputs online in SRAM akin to FlashAttention, and a design using dynamically weighted parallel sub-networks to maintain a balanced ratio between intermediate and head dimensions. Validated on models from 128M to 1.3B parameters, FlashMHF consistently improves perplexity and downstream task accuracy over SwiGLU FFNs, while reducing peak memory usage by 3-5x and accelerating inference by up to 1.08x. Our work establishes the multi-head design as a superior architectural principle for FFNs, presenting FlashMHF as a powerful, efficient, and scalable alternative to FFNs in Transformers.

## 1 INTRODUCTION

The Transformer architecture has become the standard for Large Language Models (LLMs) (Vaswani et al., 2017b). At its core, the Transformer block is composed of two primary components: a multi-head self-attention mechanism and a position-wise Feed-Forward Network (FFN). While multi-head attention is often credited for the model's ability to capture complex contextual relationships, the FFN module, which consumes a significant portion of the model's parameters and computation, is equally critical for its expressive power (Gerber, 2025).

Recent studies have revealed a structural symmetry between the Feed-Forward Network (FFN) and single-head attention (Geva et al., 2020), as illustrated in Figure 1. An FFN, defined as $\text{FFN}(\mathbf{X}) = \sigma(\mathbf{X}\mathbf{W_1}^\top)\mathbf{W_2}$ can be reinterpreted as $\mathbf{X}$ attending over $\mathbf{W_1}$ to retrieve values from $\mathbf{W_2}$. This formulation mirrors the core attention mechanism, $\text{Attention}(Q, K, V) = \text{softmax}\left(\frac{QK^T}{\sqrt{d_k}}\right)V$ with the primary distinction being the FFN's element-wise activation ($\sigma$) versus attention's row-wise softmax. The established success of the multi-head design in attention—which enables joint information processing from diverse representational subspaces—provides a strong rationale for investigating a similar multi-head decomposition for FFNs. Indeed, this direction was explored by previous

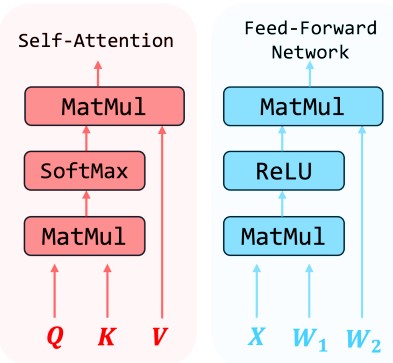

Figure 1: Structural Symmetry.

works like Multi-Head Mixture of Experts (MH-MoE) (Wu et al., 2024) for its effectiveness. However, despite its promising results, it still faces issues of scalability and low computational efficiency.

In this work, we analyze the Multi-Head Feed-Forward Network (a straightforward application of this multi-head principle), and identify two challenges that hinder its practical adoption. First, **high memory pressure**: analogous to multi-head attention, the architecture generates $H$ separate inter-

mediate activations, leading to an $H$-fold increase in memory usage—a known challenge for multi-head designs like MH-MoE. Second, **scaling imbalance**: As models scale, the FFN's intermediate dimension $d_{\text{ff}}$ grows while the per-head dimension $d_h$ remains a small constant, following the design of Multi-Head Attention. This creates a skewed $d_{\text{ff}}/d_h$ ratio degrading performance, as FFNs are known to perform optimally only when this ratio is kept within a range (Kaplan et al., 2020).

To address these challenges, we propose **FlashMHF**, whose key innovation lies in a scale-balanced structure and memory-efficient flash kernel. We partition the intermediate dimension into multiple parallel sub-networks and aggregate their outputs. This design ensures the ratio between the effective intermediate dimension and the head dimension remains balanced, maintaining performance at large scale. For memory efficiency, and analogously to FlashAttention's online softmax (Dao et al., 2022), our fused kernel computes the SwiGLU activation without materializing the large intermediate hidden state in HBM. This approach drastically reduces peak memory usage and eliminates costly data transfers between on-chip SRAM and HBM.

Extensive experiments on models from 128M to 1.3B parameters show that FlashMHF consistently outperforms the standard SwiGLU FFN baseline across all critical metrics. Our method achieves lower perplexity, stronger downstream task performance, a **1.00x-1.08x** inference speedup on the Hopper architecture, and a drastic **3-5x** reduction in peak GPU memory compared to SwiGLU FFN.

Our contributions can be summarized as follows:

- **Identification of Foundational MH-FFN Challenges.** We identify and analyze two critical issues that render a naïve Multi-Head FFN impractical: (1) high memory pressure caused by intermediate activations and (2) an architectural scaling imbalance between head and intermediate dimensions.
- **FlashMHF: A Novel and Efficient Architecture.** We propose FlashMHF, a novel architecture that resolves these challenges by pairing a scale-balanced parallel FFN sub-networks design with a high-efficiency, IO-aware kernel.
- **State-of-the-Art Performance and Efficiency.** Through extensive experiments, we demonstrate that FlashMHF significantly outperforms the widely-used SwiGLU FFN baseline in perplexity and downstream tasks, all while delivering up to a **1.08x** speedup and reducing peak memory usage by **3-5x** compared to SwiGLU FFN.

## 2 PRELIMINARIES

**Notation.** Let $L$ be the sequence length, $d_{\text{model}}$ be model dimension, $d_{\text{k}}$ be attention head dimension, and $d_{\text{ff}}$ be intermediate dimension of FFN. We consider the parameters:

$$\mathbf{Q}_{att}, \mathbf{K}_{att}, \mathbf{V}_{att} \in \mathbb{R}^{L \times d_k}, \quad \mathbf{X} \in \mathbb{R}^{L \times d_{\text{model}}}, \quad \mathbf{W}_1, \mathbf{W}_2 \in \mathbb{R}^{d_{\text{ff}} \times d_{\text{model}}}$$

For FFNs we write $\phi(\cdot)$ for an element-wise nonlinearity (e.g., ReLU, GeLU, SiLU).

**Single-Head Attention vs. FFN.**

$$\text{Att}\big(\mathbf{Q}_{att}, \mathbf{K}_{att}, \mathbf{V}_{att}\big) =: \text{softmax}\Big(\tfrac{\mathbf{Q}_{att}\mathbf{K}_{att}^{\top}}{\sqrt{d_k}}\Big)\mathbf{V}_{att}, \quad \text{FFN}(\mathbf{X}) =: \phi\big(\mathbf{X}\mathbf{W}_1^{\top}\big)\mathbf{W}_2. \tag{1}$$

By replacing the activation function $\text{softmax}(\,\cdot\,/\sqrt{d_k})$ with an element-wise nonlinearity $\phi(\cdot)$, Attention and FFN become structurally identical. Thus, we can reinterpret FFNs as "attention over parameters" of length $d_{\text{ff}}$ (Vaswani et al., 2017a; Geva et al., 2020).

$\widetilde{\text{FFN}}$ **Definition.** In modern Transformers, the gated variant $\text{SwiGLU}(\cdot)$ is the common choice instead of vanilla $\text{FFN}(\cdot)$. We follow the standard $\text{SwiGLU}(\cdot)$ formulation:

$$\text{SwiGLU}(\mathbf{X}) =: \big((\mathbf{X}\mathbf{W}_{\text{up}}) \odot \text{SiLU}(\mathbf{X}\mathbf{W}_{\text{gate}})\big)\mathbf{W}_{\text{down}}. \tag{2}$$

For later analysis, we introduce a key-value style formulation whose symbols intentionally echo attention. For any input *query-like* matrix $\mathbf{Q} \in \mathbb{R}^{L \times d}$ and $\mathbf{K}, \mathbf{U}, \mathbf{V} \in \mathbb{R}^{d_{\text{ff}} \times d}$ define

$$\widetilde{\text{FFN}}(\mathbf{Q}; \mathbf{K}, \mathbf{U}, \mathbf{V}) =: \big(\text{SiLU}(\mathbf{Q}\,\mathbf{K}^{\top}) \;\odot\; (\mathbf{Q}\,\mathbf{U}^{\top})\big)\mathbf{V}. \tag{3}$$

Under the assignments $\mathbf{Q} = \mathbf{X}$, $\mathbf{K} = \mathbf{W}_{\text{gate}}^{\top}$, $\mathbf{U} = \mathbf{W}_{\text{up}}^{\top}$, and $\mathbf{V} = \mathbf{W}_{\text{down}}$, $\widetilde{\text{FFN}}(\cdot)$ is identical to SwiGLU$(\cdot)$. Compared to vanilla FFN$(\cdot)$, SwiGLU$(\cdot)$ inserts a multiplicative gate and retains the attention-like structure since we can define a new nonlinearity function $\phi_s(\cdot)$:

$$\phi_s(\mathbf{Q}, \mathbf{K}) =: \text{SiLU}\big(\mathbf{Q}\,\mathbf{K}^{(\text{g})\top}\big) \odot \mathbf{Q}\,\mathbf{K}^{(\text{u})\top} \tag{4}$$

where $\mathbf{K}^{(\text{g})}, \mathbf{K}^{(\text{u})} \in \mathbb{R}^{d_{\text{ff}} \times d_{\text{model}}}$ and $\mathbf{K} =: [\mathbf{K}^{(\text{g})}, \mathbf{K}^{(\text{u})}] \in \mathbb{R}^{(2d_{\text{ff}}) \times d_{\text{model}}}$. Now we can rewrite SwiGLU into $\phi_s(\mathbf{Q}, \mathbf{K})\mathbf{V}$. This shows that SwiGLU is indeed a variant of attention in a broad sense, where the softmax is replaced by element-wise activation function $\phi_s(\cdot)$.

**Headwise Operation Functions.** Let $H \in \mathbb{N}$ be the number of heads and $d_h$ be the per-head dimension s.t. $d_{\text{model}} = H \cdot d_h$. For any $\mathbf{T} \in \mathbb{R}^{L \times d_{\text{model}}}$, define headwise split as

$$\text{split}_H(\mathbf{T}) \in \mathbb{R}^{L \times H \times d_h}, \qquad \big[\text{split}_H(\mathbf{T})\big]_{l,h,j} = \mathbf{T}_{l,\,(h-1)d_h+j}, \tag{5}$$

where $l \in \{1, \ldots, L\}$, $h \in \{1, \ldots, H\}$, and $j \in \{1, \ldots, d_h\}$. Simply speaking, this operation splits a tensor along the $d_{\text{model}}$ dimension into $d_h \times H$. Conversely, for any $H$ sub parts of equal sizes $\mathbf{S} \in \mathbb{R}^{L \times H \times d_h}$, define headwise concatenation

$$\text{concat}_H(\mathbf{S}) \in \mathbb{R}^{L \times d_{\text{model}}}, \qquad \big[\text{concat}_H(\mathbf{S})\big]_{l,\,(h-1)d_h+j} = \mathbf{S}_{l,h,j}. \tag{6}$$

## 3 METHODOLOGY

Our work is motivated by the structural symmetry between self-attention and FFNs, as detailed in Section 2. We posit that just as self-attention benefits from a multi-head design, the FFN can be similarly decomposed to enhance its expressive power. In this section, we first formalize the concept of a Naïve Multi-Head Feed-Forward Network (MH-FFN), discuss its practical limitations, and then introduce our proposed Flash Multi-Head FFN (FlashMHF) architecture, which leverages a gated aggregation of parallel sub-networks and flash algorithms to overcome these practical limitations.

### 3.1 NAÏVE MULTI-HEAD FEED-FORWARD NETWORKS

**Setup.** Let $H$ be the number of heads of MH-FFN, and $d_h = d_{\text{model}}/H$ be the per-head dimension.

**Definition.** Given $\mathbf{X} \in \mathbb{R}^{L \times d_{\text{model}}}$, form per-head inputs by linear projection and reshaping:

$$\mathbf{Q} = \text{split}_H\big(\mathbf{X}\,\mathbf{W}_{\text{in}}\big) \in \mathbb{R}^{L \times H \times d_h}, \qquad \mathbf{W}_{\text{in}} \in \mathbb{R}^{d_{\text{model}} \times d_{\text{model}}}. \tag{7}$$

Define matrix $\mathbf{K}^h, \mathbf{U}^h, \mathbf{V}^h \in \mathbb{R}^{d_{\text{ff}} \times d_h}$ for all heads $h \in \{1, \ldots, H\}$. Apply $\widetilde{\text{FFN}}(\cdot)$ defined in equation 3 in a headwise manner:

$$\mathbf{S}_{:,h,:} = \widetilde{\text{FFN}}\big(\mathbf{Q}_{:,h,:};\, \mathbf{K}^h, \mathbf{U}^h, \mathbf{V}^h\big) \in \mathbb{R}^{L \times d_h}. \tag{8}$$

Finally, $\mathbf{S}$ is concatenated and linearly projected to combine information across all heads:

$$\mathbf{O}^{\text{MH-FFN}} = \text{concat}_H(\mathbf{S})\,\mathbf{W}_{\text{out}} \in \mathbb{R}^{L \times d_{\text{model}}}, \qquad \mathbf{W}_{\text{out}} \in \mathbb{R}^{d_{\text{model}} \times d_{\text{model}}}. \tag{9}$$

**Limitations.** This naïve design suffers from two main limitations. The first is **Scaling Failure**: Although the multi-head structure offers expressiveness gains at smaller model scales, we empirically find that this approach ceases to be competitive once the model size exceeds $\approx 128\text{M}$ parameters, as reported in Section 4.1. The second limitation is **Memory Pressure**. MH-FFN materializes $H$ separate sets of intermediate activations, each of size approximately $L \times d_{\text{ff}}$. This incurs a total activation memory footprint of $\mathcal{O}((L \cdot H + d_{\text{model}}) \cdot d_{\text{ff}})$. Memory usage therefore grows linearly with the number of heads $H$, which quickly becomes prohibitive as models and context lengths scale.

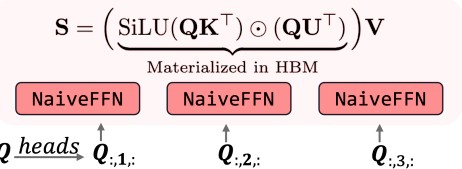

$$\mathbf{S} = \Big( \underbrace{\text{SiLU}(\mathbf{Q}\mathbf{K}^{\top}) \odot (\mathbf{Q}\mathbf{U}^{\top})}_{\text{Materialized in HBM}} \Big) \mathbf{V}$$

Figure 2: Memory limitation of MH-FFN.

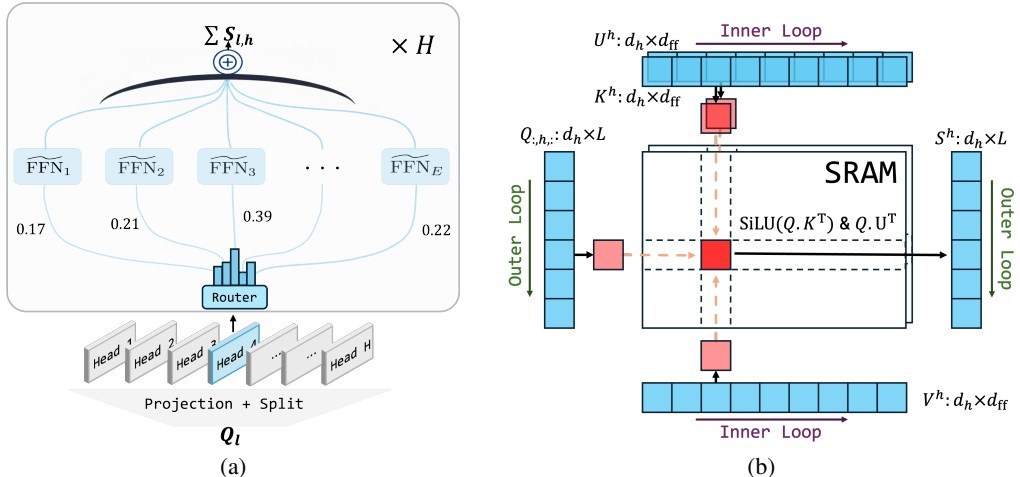

Figure 3: (a) Parallel FFN Sub-Networks. (b) $\widetilde{\text{SRAMFFN}}$ loads blocks of $\mathbf{Q}$ in the outer loop and blocks of $\mathbf{K}, \mathbf{U}, \mathbf{V}$ in the inner loop, compute $\text{SiLU}(\mathbf{Q}\mathbf{K}^\top)$, $\mathbf{Q}\mathbf{U}^\top$ and corresponding $\mathbf{V}$ multiplication on SRAM.

## 3.2 FLASH MULTI-HEAD FEED-FORWARD NETWORKS

### 3.2.1 PARALLEL FFN SUB-NETWORKS

To address the **Scaling Failure**, we first analyze why the naïve Multi-Head FFN fails to scale beyond $\approx 128\text{M}$ parameters. As the model size increases, the intermediate width $d_{\text{ff}}$ must grow, while the per-head width $d_h$ is typically kept fixed (e.g., $d_h = 128$), a design choice inherited from Multi-Head Attention (MHA). Consequently, the ratio $d_{\text{ff}}/d_h$ grows excessively. In the original SwiGLU design, a classical choice is $d_{\text{ff}}/d_{\text{model}} = \frac{8}{3}$. In contrast, under our naïve multi-head setting, we observe that this effective ratio explodes across scales:

$$128\text{M}: \quad \frac{d_{\text{ff}}}{d_h} = \frac{2048}{128} = 16, \qquad 370\text{M}: \quad \frac{d_{\text{ff}}}{d_h} = \frac{2688}{128} = 21, \qquad 1.3\text{B}: \quad \frac{d_{\text{ff}}}{d_h} = \frac{5760}{128} = 45.$$

Such a significant **Scaling Imbalance** from the optimal range (Kaplan et al., 2020) leads to a decline in parameter efficiency, rendering the naïve multi-head construction increasingly ineffective at larger scales. This diagnosis motivates our use of multiple, parallel FFN pathways, which are combined via a learned gating mechanism, illustrated in Figure 3a. Our architecture draws inspiration from Mixture-of-Experts (Shazeer et al., 2017), essentially functioning as a dense MoE structure that omits sparse top-k expert selection. This methodology re-establishes a balanced and effective expansion ratio for each head's computation without inflating per-head activations.

**Definition.** Given $\mathbf{X} \in \mathbb{R}^{L \times d_{\text{model}}}$, form per-head inputs by linear projection and reshaping:

$$\mathbf{Q} = \text{split}_H(\mathbf{X}\,\mathbf{W}_{\text{in}}) \in \mathbb{R}^{L \times H \times d_h}, \qquad \mathbf{W}_{\text{in}} \in \mathbb{R}^{d_{\text{model}} \times d_{\text{model}}}. \tag{10}$$

Let $E$ be the number of sub-networks and $d_e$ be the dimension per sub-network, such that the total intermediate dimension is $d_{\text{ff}} = E \cdot d_e$. We adopt the standard SwiGLU ratio by setting $d_e \approx \frac{8}{3} d_h$ (Touvron et al., 2023). For each head $h$, we define a *private* set of $E$ sub-networks, where the parameters for the $e$-th sub-network within head $h$ are $\mathbf{K}_e^h, \mathbf{U}_e^h, \mathbf{V}_e^h \in \mathbb{R}^{d_e \times d_h}$.

**Gating weights.** For each head $h$, we introduce a gating matrix $\mathbf{W}^h \in \mathbb{R}^{d_h \times E}$. The per-token logits for the $E$ sub-networks are computed by projecting the head's query:

$$\mathbf{P}^h = \mathbf{Q}_{:,h,:}\,\mathbf{W}^h \in \mathbb{R}^{L \times E}. \tag{11}$$

These logits are then transformed into normalized gating weights, $\mathbf{R}^h$, via a sigmoid activation followed by a numerically stable normalization.

$$\mathbf{R}_{\ell,e}^h = \frac{\sigma(\mathbf{P}_{\ell,e}^h)}{\sum_{e'=1}^{E} \sigma(\mathbf{P}_{\ell,e'}^h) + \varepsilon}, \qquad e = 1, \ldots, E, \tag{12}$$

**Sub-network Aggregation.** The final output for each head is computed as a weighted sum of its sub-network outputs, using the gating weights $\mathbf{R}^h$ to aggregate them:

$$\mathbf{S}_{\ell,h,:} = \sum_{e=1}^{E} \mathbf{R}_{\ell,e}^h \, \widetilde{\mathrm{FFN}}\big(\mathbf{Q}_{\ell,h,:}; \mathbf{K}_e^h, \mathbf{U}_e^h, \mathbf{V}_e^h\big) \ \in \ \mathbb{R}^{d_h}. \tag{13}$$

This parallel FFN sub-networks formulation allows each sub-network to maintain a balanced internal dimension ($d_e \approx \frac{8}{3}d_h$), thereby resolving the scaling imbalance identified in the naïve MH-FFN.

Finally, the outputs from all heads are concatenated and transformed by a final output projection:

$$\mathbf{O}^{\mathrm{FlashMHF}} \ = \ \mathrm{concat}_H(\mathbf{S}) \, \mathbf{W}_{\mathrm{out}} \ \in \ \mathbb{R}^{L \times d_{\mathrm{model}}}, \qquad \mathbf{W}_{\mathrm{out}} \in \mathbb{R}^{d_{\mathrm{model}} \times d_{\mathrm{model}}}. \tag{14}$$

### 3.2.2 I/O-Aware Flash Algorithm

To address the **Memory Pressure**, we introduce an I/O-aware algorithm for the FFN computation that avoids materializing the large intermediate activation tensor.

**Blockwise Computation.** Recall from equation 3 that the core computation for a single head involves an input query $\mathbf{Q} \in \mathbb{R}^{L \times d_h}$ and parameter matrices $\mathbf{K}, \mathbf{U}, \mathbf{V} \in \mathbb{R}^{d_{\mathrm{ff}} \times d_h}$. The naïve approach would compute the full intermediate tensor $\mathbf{A} = \mathrm{SiLU}(\mathbf{Q}\mathbf{K}^\top) \odot (\mathbf{Q}\mathbf{U}^\top)$.

Our flash algorithm, illustrated in Figure 3b, circumvents this by processing the computation in blocks. We partition the parameter matrices $\mathbf{K}, \mathbf{U}$, and $\mathbf{V}$ along their first dimension ($d_{\mathrm{ff}}$) into $M$ blocks of size $b$, denoted as $\{\mathbf{K}_m, \mathbf{U}_m, \mathbf{V}_m\}_{m=1}^M$, where $d_{\mathrm{ff}} = M \cdot b$. The final output $\mathbf{O} \in \mathbb{R}^{L \times d_h}$ is then computed iteratively, accumulating the result of each block one at a time. This entire loop is executed within a single fused kernel:

$$\mathbf{O} \leftarrow \mathbf{0}; \qquad \text{for } m = 1 \ldots M: \quad \mathbf{O} \leftarrow \mathbf{O} + \big(\mathrm{SiLU}(\mathbf{Q}\mathbf{K}_m^\top) \odot (\mathbf{Q}\mathbf{U}_m^\top)\big)\mathbf{V}_m. \tag{15}$$

The key to solving the high memory pressure lies in the multi-head design itself. A naïve implementation would materialize $H$ large intermediate tensors in HBM. Our algorithm avoids this entirely by leveraging the narrow heads to process the computation in blocks along the $d_{\mathrm{ff}}$ dimension, each fitting within on-chip SRAM. This blockwise execution principle also contrasts with standard FFNs, which must materialize their single, large intermediate tensor before the final projection. By design, our I/O-aware algorithm resolves the memory pressure of the naïve approach, reducing consumption from $\mathcal{O}((d_{\mathrm{ff}} \cdot H + d_{\mathrm{model}}) \cdot L)$ to $\mathcal{O}(d_{\mathrm{model}} \cdot L)$. Remarkably, this memory footprint is even smaller than that of a standard SwiGLU FFN, which requires $\mathcal{O}((d_{\mathrm{ff}} + d_{\mathrm{model}}) \cdot L)$. Detailed pseudocode for the forward and backward passes is provided in Appendices A and B.

## 4 Experiments

For a fair comparison, all models are pre-trained with a context length of $4{,}096$ and a batch size of $64$. We ensure that the sizes of our models were approximately equal to those of the baselines, and we synchronize the hyper-parameter settings for the optimizer across all models.

**Baseline.** Our baseline is a Llama-Like (Touvron et al., 2023) model consisting of multi-head self-attention with Rotary Position Embeddings (RoPE) (Su et al., 2021) and a point-wise SwiGLU FFN. We use the GPT-NeoX (Black et al., 2022) tokenizer with a vocabulary size of $50{,}432$. We disable attention dropout and FFN bias throughout. RoPE uses $\theta{=}10{,}000$, and the maximum position embedding length is set to $4{,}096$. We trained baseline as well as our models across 128M, 370M and 1.3B to fully validate scalability of our model. In terms of other configurations, we generally follow the same settings and hyperparameters from the appendix of Dao & Gu (2024).

**Parametric KV Baseline.** As a key ablation study, we introduce the PKV baseline, which replaces the SwiGLU FFN with a multi-head attention whose keys and values are learnable model parameters. This baseline approximates an extreme case of our architecture where each sub-network has a dimension of one ($d_e{=}1$) and serves to validate the necessity of the point-wise SwiGLU component. For fair comparison, its parameter size is matched to the primary baseline.

**Dense-MoE Baseline.** To verify whether the gains come from the multi-head design or from the parallel FFN sub-networks, we set the number of heads $H$ to 1. This control group is equivalent to a dense MoE.

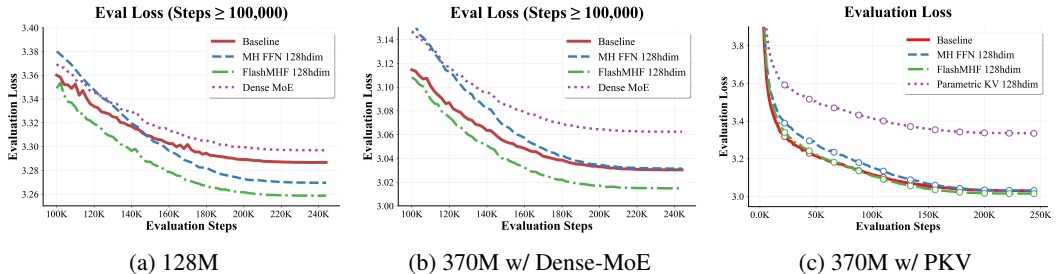

| (a) 128M | (b) 370M w/ Dense-MoE | (c) 370M w/ PKV |
|---|---|---|

Figure 4: Comparing Baseline, Parametric KV, FlashMHF and MH-FFN in 128M and 370M scales.

**MH-FFN.** We replace the baseline point-wise SwiGLU with our MH-FFN module. We adjust the layer depth and intermediate width to achieve approximately equal model size as the baseline. For fair comparison, we set the model size and FLOPs approximately equal to the baseline by adjusting the layer depth and intermediate width. More detailed configs per model scale are listed in Appendix D.

**FlashMHF.** We replace point-wise SWIGLU FFN in the baseline model with our FlashMHF module, we keep almost all components identical to the baseline (attention configuration, layernorms, dropout $= 0$, and no FFN biases). We experiment with FlashMHF of different FFN head dimensions across $d_h = \{64, 128, 256\}$. We set the dimension per sub-network to $d_e \approx \frac{8}{3} d_h$ rounded up to the nearest multiple of 64 and set $E$ accordingly. We adjust the layer depth and intermediate width to achieve approximately equal model size as the baseline.

**Data and training tokens.** All models are trained on THE PILE. We train the 128M and 370M models with 60B tokens (245K steps), and train the 1.3B model with 100B tokens (409K steps). We calculate evaluation loss on PG19's validation split. Further optimizer details, learning-rate schedules, regularization, and all remaining hyperparameters are provided in Appendix C.

### 4.1 LANGUAGE MODELING AT DIFFERENT SCALES

**State-of-the-Art Performance.** We first experiment with the SwiGLU baseline, Dense-MoE baseline, PKV-128hdim and FlashMHF-128hdim in 128M and 370M scale.

The validation loss are presented in Figure 4 and Table 1. The results clearly demonstrate the superiority of our approach in both scales. FlashMHF consistently achieves a lower final validation loss than the strong SwiGLU baseline, while the PKV baseline performs notably poorly in comparison.

The superior performance of FlashMHF stems from fundamental advantages in its architectural design. We analyze these through two comparisons: First, against the SwiGLU FFN, FlashMHF's multi-head design significantly enhances expressive power. We hypothesize this advantage can be understood through the lens of "implicit thinking", where a standard FFN is viewed as executing a single path of sequential reasoning (Chen et al., 2025). In this framework, FlashMHF's architecture is analogous to performing a *beam search* over this implicit thinking process. By exploring multiple

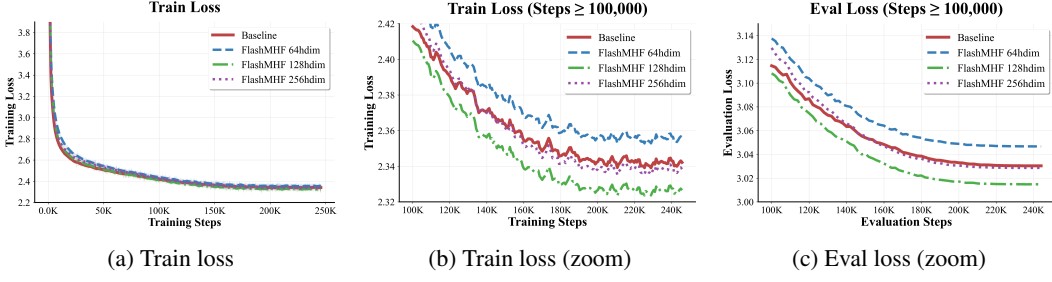

| (a) Train loss | (b) Train loss (zoom) | (c) Eval loss (zoom) |
|---|---|---|

Figure 5: Training on 370M model scale to investigate the best head dimension. Analysis: (a) is full training loss, to visualize it more clearly, we zoom in to later training steps as illustrated in (b) and (c). Our FlashMHF with $d_h = 64, 128$ gets better train/evaluation loss on PG19 validation split.

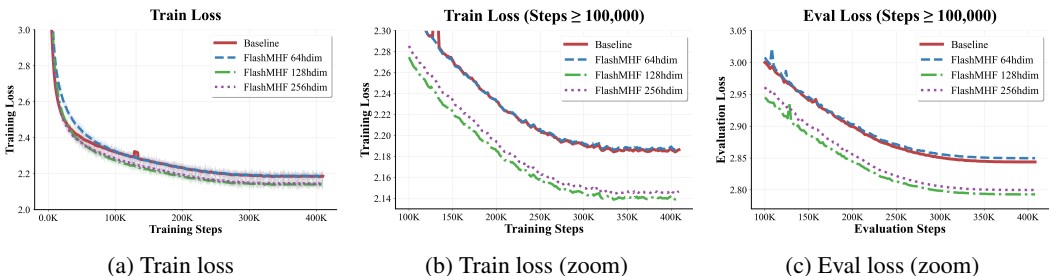

| (a) Train loss | (b) Train loss (zoom) | (c) Eval loss (zoom) |

Figure 6: Scaling FlashMHF up to 1.3B. FlashMHF with $d_h = 128$ constantly performs the best.

cognitive paths in parallel, the model can construct richer and more robust representations, naturally boosting its capabilities. Second, when contrasted with the PKV baseline, the utility of element-wise structure is clear. PKV's softmax activation induces competition across the entire hidden dimension, creating an aggressive information bottleneck. In contrast, element-wise activation function avoids this competitive pressure, maximizing parameter efficiency. This allows each channel to learn more freely, enabling the model to form a much richer and more disentangled set of features, which ultimately leads to better performance. Meanwhile, we observe that Dense-MoE even underperforms the baseline. We attribute this to the ratio of the intermediate dimension to the input dimension deviating from its optimal value, further supporting our FFN ratio hypothesis.

**Ablation on Parallel FFN Sub-Networks.** At the 128M scale (Figure 4a), both MH-FFN-128hdim and FlashMHF-128hdim outperform the SwiGLU baseline on the PG19 validation set. However, a clear divergence emerges at the 370M scale (Table 1, Figure 4b): the naïve MH-FFN is no longer competitive, while FlashMHF continues to deliver gains. Given that the sole mathematical difference between these models is our parallel FFN sub-networks, this result empirically validates it as the crucial component for successful scaling.

The performance divergence of MH-FFN across these model scales provides a crucial insight that supports our arguments in Section 3.2. The fact that MH-FFN is effective at 128M but fails to scale to 370M is direct evidence for our reasoning: as model size grows, the ratios $d_{\text{ff}}/d_h$ of MH-FFN become imbalanced, inevitably leading to performance loss. This strongly validates that our use of parallel sub-networks, which introduces multiple smaller FFN pathways in FlashMHF, is a well-calibrated and critical solution addressing this scaling challenge.

**Head-Dimension Ablation.** Building on the those findings, we now probe how the FlashMHF performance depends on its head dimension. At the 370M scale, we vary the FlashMHF head dimension $d_h \in \{64, 128, 256\}$ as illustrated in Figure 5 and shown in Table 1. FlashMHF with $d_h = 128$ and 256 outperforms the SwiGLU Baseline; moreover, $d_h = 128$ offers the best performance with a 0.015 margin in evaluation loss, whereas $d_h = 64$ underfits and $d_h = 256$ yields diminishing returns.

These results indicate that moderate head dimensions are usually more preferable and point to a fundamental trade-off between per-head expressive power and the architectural benefit of subspace diversity, which is similar to the conclusions in Wu et al. (2024). A small dimension such as $d_h = 64$ creates a representational bottleneck in each head, leading to underfitting as individual pathways lack the capacity to learn complex features. Conversely, a large dimension like $d_h = 256$ reduces the total number of heads, diminishing the gains from functional specialization and causing the

Table 1: Evaluation loss at 370M and 1.3B scales.

| Model Size 370M | Loss | Model Size 1.3B | Loss |
|---|---|---|---|
| Baseline | 3.030 | Baseline | 2.843 |
| PKV ($d_h$=128) | 3.334 | – | – |
| MH-FFN ($d_h$=128) | 3.031 | – | – |
| Dense-MoE | 3.062 | – | – |
| FlashMHF ($d_h$=64) | 3.046 | FlashMHF ($d_h$=64) | 2.849 |
| FlashMHF ($d_h$=128) | **3.014** | FlashMHF ($d_h$=128) | **2.793** |
| FlashMHF ($d_h$=256) | 3.029 | FlashMHF ($d_h$=256) | 2.799 |

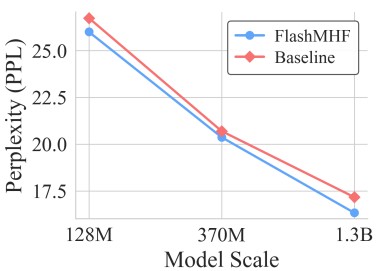

Figure 7: PPL vs Model Scale.

| Scale | Model | HellaSwag↑ | SIQA↑ | PIQA↑ | OBQA↑ | WinoGrande↑ | RACE↑ | Average↑ |
|-------|-------|-----------|-------|-------|-------|-------------|-------|----------|
| 370M | Baseline | 33.20 | 40.94 | 64.53 | 27.50 | 51.70 | 21.66 | 39.92 |
| 370M | Parametric KV | 27.85 | 39.61 | 60.50 | 27.20 | 51.78 | 21.67 | 38.10 |
| 370M | MH-FFN | 33.45 | 41.30 | 65.18 | 27.60 | 51.46 | 21.87 | 40.14 |
| 370M | FlashMHF-64hdim | 32.57 | **41.50** | 65.34 | 27.60 | **53.35** | 21.59 | 40.32 |
| 370M | FlashMHF-128hdim | **33.97** | 40.63 | **66.27** | **27.80** | 52.41 | 21.80 | **40.48** |
| 370M | FlashMHF-256hdim | 33.60 | 39.71 | 64.85 | 27.30 | 52.17 | **21.94** | 39.92 |
| 1.3B | Baseline | 39.47 | 41.76 | 67.30 | 27.60 | 52.64 | 21.73 | 41.75 |
| 1.3B | FlashMHF-64hdim | 39.80 | 42.17 | 67.08 | 27.20 | 51.30 | **22.62** | 41.70 |
| 1.3B | FlashMHF-128hdim | **42.96** | **44.17** | 68.44 | **27.80** | **54.46** | 22.26 | **43.35** |
| 1.3B | FlashMHF-256hdim | 41.80 | 42.32 | **68.88** | 27.60 | 53.35 | 22.01 | 42.66 |

Table 2: Downstream benchmarks for 1.3B and 370M models. **Bold** = best, underline = second best.

architecture to behave more like a monolithic FFN. The $d_h = 128$ configuration appears to strike an effective balance, endowing each head with sufficient capacity while maintaining a high degree of parallelism and diversity.

**Scalability.** We conducted large-scale experiments to demonstrate that the FlashMHF architecture is fundamentally scalable and that our key design principles generalize to larger models. We scaled our model to 1.3B parameters and replicated our analysis of head-dimension $d_h \in \{64, 128, 256\}$. The results, presented in Figure 6 and Table 1, lead to two critical conclusions. First, the superiority of FlashMHF over the baseline is not only preserved but even amplified at the 1.3B scale as shown in Figure 7; it converges faster and achieves a substantially lower validation loss, leading to a larger improvement of $0.85$ in perplexity. Second, our earlier findings from the 370M model that the optimal head dimension $d_h = 128$ provides significant benefits hold true at this larger scale. The consistent behavior across a nearly 4x increase in model size robustly demonstrates that the convergence superiority and performance gains of FlashMHF are a general property of the architecture, making it a viable and scalable solution for training state-of-the-art language models.

## 4.2 Downstream Task Performance

**Setup.** To ascertain whether the improved validation loss of FlashMHF translates into enhanced downstream capabilities, we evaluated our 370M and 1.3B models on a comprehensive suite of benchmarks. We assess commonsense reasoning using HellaSwag (Zellers et al., 2019), Social IQA (SIQA) (Sap et al., 2019), Physical IQA (PIQA) (Bisk et al., 2019), OpenBookQA (OBQA) (Mihaylov et al., 2018), and WinoGrande (Levesque et al., 2011; Sakaguchi et al., 2019). Additionally, we evaluate reading comprehension using the RACE dataset (Lai et al., 2017).

**Results.** As summarized in Table 2, we first note that given our modest training scale (60-100B tokens), a detailed analysis of performance on individual tasks may be of limited significance. However, we posit that the average performance across a diverse benchmark suite is a statistically robust indicator of an architecture's general capabilities, with a superior design expected to yield a consistently better average score. The results strongly support this view. Notably, as highlighted by the gray background in the table, the best performance on every individual benchmark across both the 370M and 1.3B scales is invariably achieved by a FlashMHF variant. While **FlashMHF-128hdim** secures the highest *average* score, confirming $d_h = 128$ as an effective sweet spot, the FlashMHF architecture as a whole consistently outperforms the SwiGLU baseline across all configurations.

## 4.3 Speed and Memory Efficiency

A core motivation for FlashMHF is to enhance model capabilities while also improving computational efficiency. To conduct a fair evaluation, we benchmarked our module against the SwiGLU FFN baseline. For latency evaluation, to ensure a comparable total parameter count, we compared a 20-layer FlashMHF (and MH-FFN) against a 24-layer SwiGLU baseline. For memory profiling, we compared the modules directly at the single-layer level. The naïve MH-FFN, included for completeness, was substantially less efficient in both metrics; our analysis therefore focuses on the

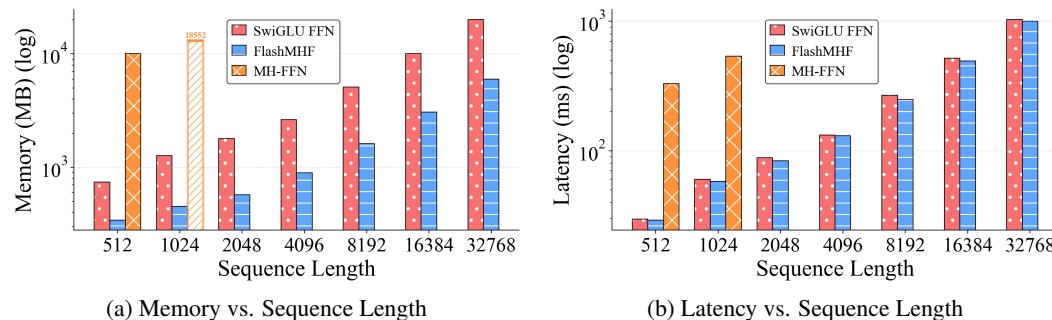

(a) Memory vs. Sequence Length          (b) Latency vs. Sequence Length

Figure 8: Memory and latency comparison of SwiGLU FFN, MH-FFN, and FlashMHF. (Log-graph)

comparison between FlashMHF and the strong SwiGLU baseline. All benchmarks were run on an Nvidia H100 GPU, with detailed configurations and results available in Appendix E.

**Memory Efficiency** FlashMHF delivers a leap in memory efficiency. As shown in Figure 8a, our I/O-aware kernel drastically reduces peak memory consumption by a factor of **3-5x** compared to a standard SwiGLU FFN. This dramatic reduction directly enables inference and training with significantly longer context lengths or deploying larger models on the same hardware. This advantage further widens as sequence length increases, underscoring the superior scalability of our design.

**Latency** Regarding latency, FlashMHF also provides a speedup, though the improvement is more moderate compared to the dramatic memory gains. Our benchmarks demonstrate a peak inference speedup of **1.08x**, with an average improvement of approximately **1.05x** across various configurations (Figure 8b). This speedup primarily stems from eliminating the I/O bottleneck of writing and reading the large intermediate activation tensor to and from HBM. It is worth noting that this latency improvement is more moderate than that of FlashAttention. The reason is that the standard FFN layer gets highly optimized by cuBLAS and has a higher GPU Memory cache hit rate. Nevertheless, this efficiency gain, while modest, is a welcome byproduct of our memory-optimized design, making FlashMHF a practical and beneficial replacement for standard FFNs in production environments.

**Summary of Findings.** Combining these efficiency results with the model quality evaluations from Section 4.2 presents a powerful conclusion. The full Transformer model equipped with FlashMHF not only achieves lower perplexity and superior downstream performance but also runs faster and consumes significantly less memory. Our analysis provides comprehensive evidence that FlashMHF offers *a rare "free lunch"*: a direct replacement for standard SwiGLU FFNs that improves every critical metric—model quality, inference speed, and memory footprint—without any trade-offs. This makes it a highly practical and scalable solution for developing next-generation language models.

## 5 RELATED WORKS

**The Structural Symmetry of FFN and Attention.** The structural symmetry between Feed-Forward Networks (FFNs) and self-attention is well-established. FFNs have been interpreted as a form of attention over their own parameters (Vaswani et al., 2017a) and more formally as distributed key-value memories that map learned patterns to output distributions (Geva et al., 2020). Our work operationalizes this symmetry to architecturally enhance the FFN itself. Recent architectures have implicitly leveraged this duality to rethink model design. MLP-Mixer (Tolstikhin et al., 2021) validates this structural duality by demonstrating that token-mixing and feature-mixing can be treated as symmetric operations via simple transposition. DaViT (Ding et al., 2022) further bridges this gap by applying self-attention mechanisms directly along the feature dimension, thereby operationalizing the symmetry between spatial and channel interactions. Most recently, Tokenformer (Wang et al., 2024) takes a more radical step by replacing all linear projections in the network with Token-Parameter Attention (PAttention). However, a direct comparison between our targeted approach and these holistic architectures is challenging due to significant structural divergence. Specifically, this divergence is most pronounced in Tokenformer, as it modifies the entire Transformer block by replacing projections in both the Attention and FFN modules with complex non-linear transformations. In contrast, FlashMHF strictly modifies the FFN layer while preserving the standard Attention mechanism.

**Multi-Head Structures in FFN-like Networks.** While prior work like Multi-Head MoE (MH-MoE) (Wu et al., 2024) explored parallel FFN heads, it faced critical scalability and expressiveness limitations. Its memory footprint grew linearly with the number of heads, and all heads shared the same expert parameters. FlashMHF overcomes both issues: our novel kernel eliminates the memory cost of intermediate activations, and it enables each head to learn independent parameters, thereby unlocking the full potential of the multi-head FFN architecture. Due to the limitation that all head parameters are shared in MH-MoE, it requires $H$ times more FLOPS than our method for the same number of parameters in dense mode, making fair comparison with FlashMHF infeasible.

**IO-Aware Kernel Design.** Our work is inspired by the I/O-aware kernel design paradigm popularized by the FlashAttention series (Dao et al., 2022; Dao, 2023; Shah et al., 2024). This approach mitigates memory bandwidth bottlenecks by fusing operations and using fast on-chip SRAM to avoid materializing large intermediate tensors in slower HBM. We apply this principle to the multi-head FFN, developing a custom kernel that circumvents the need to store large intermediate activations. For modern hardware like the NVIDIA Hopper architecture, we further incorporate techniques such as asynchronous data movement and warp-group specialization to maximize throughput.

# 6 CONCLUSION

In this work, we introduce FlashMHF, a novel multi-head FFN architecture designed for superior performance and computational efficiency. By uniting a scale-balanced parallel FFN sub-networks design with a memory-frugal I/O-aware kernel, FlashMHF delivers a rare trifecta of gains: it decisively outperforms the strong SwiGLU baseline on perplexity and downstream tasks, while slashing peak memory by 3-5x and accelerating inference. We establish FlashMHF as a fundamentally superior and scalable successor to standard FFNs, setting a new state-of-the-art for dense model architectures and charting a clear course for more powerful and efficient language models.

# 7 REPRODUCIBILITY STATEMENT

To ensure the reproducibility of our results, we will make all associated code publicly available, including the implementation of the FlashMHF I/O-aware kernel, model configurations, and training scripts. The code will be released under an open-source license at the following URL: `https://anonymous.4open.science/r/FlashMHF-9395`.

All of our models were trained on THE PILE, which is a publicly available dataset. Our experimental setup, including detailed hyperparameters, optimizer settings, and specific model configurations for each scale, is described in Appendix D and C. Experiments were conducted using PyTorch on NVIDIA H100 GPUs. Upon publication, we also intend to release the final model checkpoints for our 1.3B variants to facilitate future research.

# 8 ETHICS STATEMENT

This research introduces FlashMHF, a new and more efficient architecture for language models. As a foundational work, it does not propose a specific downstream application. The primary ethical considerations are therefore those inherent to the development of all Large Language Models. These include the potential for models built with this architecture to generate false, misleading, or harmful content, and the risk of perpetuating societal biases present in the training data.

At the same time, our work offers positive ethical implications. By drastically reducing the memory footprint and improving the computational efficiency of inference, FlashMHF contributes to making powerful AI more sustainable and accessible. The reduced hardware requirements can lower the energy consumption associated with LLMs and help democratize access to this technology. This enables a wider range of researchers and organizations to develop, study, and align these models in a safe and responsible manner.

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

## A    TRITON PSEUDO CODE FOR SRAMFFN

Triton code can run efficiently on consumer grade GPUs (like RTX3090) but not on Hopper cards.

---

**Algorithm 1** SRAMFFN-FORWARD-TRITON

---

**Require:** $Q \in \mathbb{R}^{B \times H \times L \times d_h}$, $K, U, V \in \mathbb{R}^{H \times E \times d_e \times d_h}$, $R \in \mathbb{R}^{B \times H \times L \times E}$

    **Strategy:** Parallelize over batch, head, and sequence blocks. Inner loops iterate over sub-networks and intermediate dimension blocks.

    **Meta:** BLOCK_SEQ, BLOCK_INTER, HEAD_DIM= $d_h$, PFFN_DIM= $d_e$, NUM_PFFN= $E$

    **Grid:** pid0←seq_block, pid1←head $h$, pid2←batch $b$

1: $s_0 \leftarrow$ pid0·BLOCK_SEQ
2: $Q_{\text{blk}} \leftarrow Q[b, h, s_0 : s_0+\text{BLOCK\_SEQ}, :]$                 ▷ masked at sequence tail
3: $O_{\text{acc}} \leftarrow \mathbf{0}^{\text{BLOCK\_SEQ} \times d_h}$                                  ▷ SRAM accumulator
4: **for** $e = 0$ **to** $E-1$ **do**
5:      $R_{\text{rows}} \leftarrow R[b, h, s_0 : s_0+\text{BLOCK\_SEQ}, e]$
6:      **for** $m = 0$ **to** $d_e-1$ **step** BLOCK_INTER **do**
7:          $K_{\text{tile}} \leftarrow K[h, e, m{:}m+\text{BLOCK\_INTER}, :]^{\top}$
8:          $U_{\text{tile}} \leftarrow U[h, e, m{:}m+\text{BLOCK\_INTER}, :]^{\top}$
9:          $V_{\text{tile}} \leftarrow V[h, e, m{:}m+\text{BLOCK\_INTER}, :]$
10:         $M \leftarrow Q_{\text{blk}} \cdot K_{\text{tile}}; \quad N \leftarrow Q_{\text{blk}} \cdot U_{\text{tile}}$
11:         $A \leftarrow \text{SiLU}(M) \odot N; \quad A \leftarrow A \odot R_{\text{rows}}$
12:         $O_{\text{acc}} \leftarrow O_{\text{acc}} + A \cdot V_{\text{tile}}$
13:      **end for**
14: **end for**
15: $O[b, h, s_0 : s_0+\text{BLOCK\_SEQ}, :] \leftarrow O_{\text{acc}}$

---

---

**Algorithm 2** SRAMFFN-BACKWARD-TRITON(DQ, DR)

---

**Require:** saved $Q, K, U, V, R$; incoming $dS \in \mathbb{R}^{B \times H \times L \times d_h}$

    **Strategy:** Parallelize over batch, head, and sequence blocks. Inner loops iterate over sub-networks and intermediate dimension blocks.

    **Grid/Meta** as in Alg 1

1: $s_0 \leftarrow$ pid0·BLOCK_SEQ; $h \leftarrow$ pid1; $b \leftarrow$ pid2
2:    $Q_{\text{blk}} \leftarrow Q[b, h, s_0 : s_0+\text{BLOCK\_SEQ}, :]; \quad dS_{\text{blk}} \leftarrow dS[b, h, s_0 : s_0+\text{BLOCK\_SEQ}, :]$
3:    $dQ_{\text{acc}} \leftarrow \mathbf{0}^{\text{BLOCK\_SEQ} \times d_h}; \quad dR_{\text{rows}} \leftarrow \mathbf{0}^{\text{BLOCK\_SEQ} \times 1}$
4:    **for** $e = 0$ **to** $E-1$ **do**
5:      $R_{\text{rows}} \leftarrow R[b, h, s_0 : s_0+\text{BLOCK\_SEQ}, e]$
6:      **for** $m = 0$ **to** $d_e-1$ **step** BLOCK_INTER **do**
7:        $K_{\text{tile}} \leftarrow K[h, e, m{:}m+\text{BLOCK\_INTER}, :];$
8:        $U_{\text{tile}} \leftarrow U[h, e, m{:}m+\text{BLOCK\_INTER}, :];$
9:        $V_{\text{tile}}^{\top} \leftarrow V[h, e, m{:}m+\text{BLOCK\_INTER}, :]^{\top}$
10:       $M \leftarrow Q_{\text{blk}} \cdot K_{\text{tile}}^{\top}; \quad N \leftarrow Q_{\text{blk}} \cdot U_{\text{tile}}^{\top}$
11:       $\text{sig} \leftarrow \text{sigmoid}(M); \quad \text{SiLU}(M) \leftarrow M \odot \text{sig}$
12:       $dA \leftarrow dS_{\text{blk}} \cdot V_{\text{tile}}^{\top}$
13:       $dR_{\text{rows}} \mathrel{+}= \sum_{\text{cols}}\big(dA \odot \text{SiLU}(M) \odot N\big)$
14:       $dM \leftarrow \big(dA \odot (R_{\text{rows}} \odot N)\big) \odot \big(\text{sigmoid}(M) + M \cdot \text{sigmoid}(M) \cdot (1 - \text{sigmoid}(M))\big)$
15:       $dN \leftarrow \big(dA \odot \text{SiLU}(M)\big) \odot R_{\text{rows}}$
16:       $dQ_{\text{acc}} \mathrel{+}= dM \cdot K_{\text{tile}} + dN \cdot U_{\text{tile}}$
17:      **end for**
18:      $dR[b, h, s_0 : s_0+\text{BLOCK\_SEQ}, e] \leftarrow dR_{\text{rows}}$
19:      **end for**
20:    $dQ[b, h, s_0 : s_0+\text{BLOCK\_SEQ}, :] \leftarrow dQ_{\text{acc}}$

---

---

**Algorithm 3** SRAMFFN-Backward-Triton($dK$, $dU$, $dV$)

---

**Require:** saved $Q, K, U, V, R$; $dS$

    **Grid:** $\texttt{pid0} \leftarrow e \cdot (d_e/\texttt{BLOCK\_INTER}) + \lfloor m/\texttt{BLOCK\_INTER} \rfloor$, $\texttt{pid1} \leftarrow h$

1: decode $(e, m)$ from $\texttt{pid0}$
2: $K_{\text{tile}} \leftarrow K[h, e, m{:}m{+}\texttt{BLOCK\_INTER}, :]$; $U_{\text{tile}} \leftarrow U[h, e, m{:}m{+}\texttt{BLOCK\_INTER}, :]$; $V_{\text{tile}}^{\top} \leftarrow V[h, e, m{:}m{+}\texttt{BLOCK\_INTER}, :]^{\top}$
3: $dK_{\text{acc}}, dU_{\text{acc}}, dV_{\text{acc}} \leftarrow \mathbf{0}^{\texttt{BLOCK\_INTER} \times d_h}$
4: **for** $b = 0$ **to** $B{-}1$ **do**
5:     **for** $s_0 = 0$ **to** $L{-}1$ **step** $\texttt{BLOCK\_SEQ}$ **do**
6:         $Q_{\text{blk}} \leftarrow Q[b, h, s_0 : s_0{+}\texttt{BLOCK\_SEQ}, :]$;
7:         $R_{\text{rows}} \leftarrow R[b, h, s_0 : s_0{+}\texttt{BLOCK\_SEQ}, e]$;
8:         $dS_{\text{blk}} \leftarrow dS[b, h, s_0 : s_0{+}\texttt{BLOCK\_SEQ}, :]$
9:         $M \leftarrow Q_{\text{blk}} \cdot K_{\text{tile}}^{\top}$;   $N \leftarrow Q_{\text{blk}} \cdot U_{\text{tile}}^{\top}$
10:        $\text{sig} \leftarrow \text{sigmoid}(M)$;   $\text{SiLU}(M) \leftarrow M \odot \text{sig}$;   $\widetilde{N} \leftarrow R_{\text{rows}} \odot N$
11:        $A^{\top} \leftarrow (\text{SiLU}(M) \odot \widetilde{N})^{\top}$;   $dA \leftarrow dS_{\text{blk}} \cdot V_{\text{tile}}^{\top}$
12:        $dM \leftarrow (dA \odot \widetilde{N}) \odot \big( \text{sigmoid}(M) + M \cdot \text{sigmoid}(M) \cdot (1 - \text{sigmoid}(M)) \big)$;
13:        $dN \leftarrow (dA \odot \text{SiLU}(M)) \odot R_{\text{rows}}$
14:        $dV_{\text{acc}} \mathrel{+}= A^{\top} \cdot dS_{\text{blk}}$
15:        $dK_{\text{acc}} \mathrel{+}= dM^{\top} \cdot Q_{\text{blk}}$;   $dU_{\text{acc}} \mathrel{+}= dN^{\top} \cdot Q_{\text{blk}}$
16:     **end for**
17: **end for**
18: $dK[h, e, m{:}m{+}\texttt{BLOCK\_INTER}, :] \leftarrow dK_{\text{acc}}$;  $dU[h, e, m{:}m{+}\texttt{BLOCK\_INTER}, :] \leftarrow dU_{\text{acc}}$; $dV[h, e, m{:}m{+}\texttt{BLOCK\_INTER}, :] \leftarrow dV_{\text{acc}}$

---

# B   Hopper/ThunderKittens Pseudocode for SRAMFFN

---

**Algorithm 4** SRAMFFN-Forward-TK (Hopper)

---

**Require:** $Q \in \mathbb{R}^{B \times H \times L \times d_h}$, $K, U, V \in \mathbb{R}^{H \times E \times d_e \times d_h}$, $R \in \mathbb{R}^{B \times H \times E \times L}$

    **Meta:** $\texttt{BLOCK\_SEQ}$, $\texttt{BLOCK\_INTER}$, $\texttt{NUM\_STAGES}$, $\texttt{CON\_WARPGRPS} \geq 2$, $\texttt{PROD\_WARPGRPS} = 1$, $d_h{=}128$, $d_e \bmod \texttt{BLOCK\_INTER} = 0$

    **Grid:** $x = \lceil L/(\texttt{BLOCK\_SEQ} \cdot \texttt{CON\_WARPGRPS}) \rceil$, $y = H$, $z = B$

1: Allocate stage/ring buffers in SRAM for $Q$ (per consumer), $R$ (per consumer), and $K, U, V$ (per stage).
2: **Warmup (producer):** prefetch the $Q$ tiles for all consumers in this $x$-block; prefetch $R$ for subnet $e{=}0$; prefetch the first $\texttt{NUM\_STAGES}$ $(K, U, V)$ inter-tiles.

3: **Producer loop (over inter-tiles):**
4: **for** inter\_tile $= \texttt{NUM\_STAGES}, \ldots, E \cdot (d_e/\texttt{BLOCK\_INTER}){-}1$ **do**
5:     wait for consumers to finish the target stage; then prefetch the next $(K, U, V)$ tiles into that stage.
6:     **if** inter\_tile is the first tile of a new subnet $e$ **then**
7:         prefetch router $R$ rows for all consumers (current $x$-block).
8:     **end if**
9: **end for**

10: **Each consumer warpgroup** $c \in \{0, \ldots, \textbf{CON\_WARPGRPS}{-}1\}$ (independent, identical):
11: load its $Q$ tile; set $O_{\text{acc}} \leftarrow 0$.
12: **for** inter\_tile $= 0, \ldots, E \cdot (d_e/\texttt{BLOCK\_INTER}){-}1$ **do**
13:     wait until producer has filled the current stage with $(K, U, V)$; if this tile starts a new subnet, wait for $R$.
14:     $M \leftarrow Q_{\text{blk}} K_{\text{tile}}^{\top}$;   $N \leftarrow Q_{\text{blk}} U_{\text{tile}}^{\top}$
15:     $S \leftarrow \text{SiLU}(M) \odot N$;   $S \leftarrow S \odot r$                        ▷ apply router row $r$
16:     $O_{\text{acc}} \leftarrow O_{\text{acc}} + S\, V_{\text{tile}}$
17:     signal producer that this stage can be reused.
18: **end for**
19: store $O_{\text{acc}}$ to global output.

---

---

**Algorithm 5** SRAMFFN-BACKWARD-TK (HOPPER)

---

**Require:** saved $Q, K, U, V, R$; upstream $dO$
    **Meta:** BLOCK_SEQ=BLOCK_INTER, NUM_STAGES$= 2$, two consumer warpgroups, one producer
    **Grid:** $x = \lceil (E \cdot d_e)/(2 \cdot \text{BLOCK\_INTER}) \rceil$, $y = H$, $z = B$

1: Assign each consumer a distinct inter-tile (A/B). Allocate per-stage SRAM for $Q$, $dO$, $R$; per-consumer SRAM for its $(K, U, V)$ tile; small scratch for partial $dQ/dR$ exchange.
2: **Warmup (producer):** prefetch $(K, U, V)$ for both consumers' inter-tiles; prefetch first-stage $Q$, $dO$, and subnet-$e$ router $R$.

3: **Producer loop (over sequence tiles):**
4: **for** each sequence tile $t$ **do**
5:     wait for consumers to release stage $t \bmod 2$; prefetch $Q[b, h, t]$, $dO[b, h, t]$, and $R[b, h, e, t]$ for that stage.
6: **end for**

7: **Consumer warpgroup #0 (inter-tile A):**
8: init $dK_{\text{acc}}, dU_{\text{acc}}, dV_{\text{acc}} \leftarrow 0$.
9: **for** each sequence tile $t$ **do**
10:     wait for producer to provide $Q, dO, R$ for stage $t \bmod 2$; use preloaded $(K, U, V)$ for A.
11:     $M \leftarrow Q_{\text{blk}} K_{\text{A}}^{\top}$;    $N \leftarrow Q_{\text{blk}} U_{\text{A}}^{\top}$
12:     $A \leftarrow \text{SiLU}(M)$;    $dA \leftarrow dO_{\text{blk}} V_{\text{A}}^{\top}$
13:     $A' \leftarrow A + \text{sigmoid}(M) \cdot (1 - A)$
14:     $dR_{\text{rows}} \mathrel{+}= \text{row\_sum}(dA \odot A \odot N)$
15:     $dN \leftarrow (dA \odot A) \odot r$;    $dM \leftarrow (dA \odot (N \odot r)) \odot A'$
16:     $dQ^{(A)} \leftarrow dN\, U_{\text{A}} + dM\, K_{\text{A}}$
17:     $dK_{\text{acc}} \mathrel{+}= dM^{\top} Q_{\text{blk}}$;    $dU_{\text{acc}} \mathrel{+}= dN^{\top} Q_{\text{blk}}$
18:     $A_{\text{gated}} \leftarrow A \odot (N \odot r)$;    $dV_{\text{acc}} \mathrel{+}= A_{\text{gated}}^{\top} dO_{\text{blk}}$
19:     write $dQ^{(A)}$ and $dR_{\text{rows}}$ to a shared slot; notify peer.
20: **end for**
21: store-add $dK_{\text{acc}}, dU_{\text{acc}}, dV_{\text{acc}}$ to global.

22: **Consumer warpgroup #1 (inter-tile B):**
23: same loop with $(K, U, V)$ for B, producing $dQ^{(B)}$, $dR_{\text{rows}}^{(B)}$, and its own $dK_{\text{acc}}, dU_{\text{acc}}, dV_{\text{acc}}$.
24: at each $t$: wait for peer's $dQ^{(A)}/dR^{(A)}$, then
25:     $dQ[b, h, t] \mathrel{+}= dQ^{(A)} + dQ^{(B)}$,    $dR[b, h, e, t] \mathrel{+}= dR^{(A)} + dR^{(B)}$,
26: then free the shared slot so producer can reuse the stage.
27: store-add its $dK_{\text{acc}}, dU_{\text{acc}}, dV_{\text{acc}}$ to global.

---

## C  TRAINING HYPERPARAMETERS

All models training at particular size are trained with optimizer with hyperparameters set in Table 3:

Table 3: Training hyperparameters by model scale.

|  | **128M** | **370M** | **1.3B** |
|---|---|---|---|
| Learning rate | 3e-3 | 1.5e-3 | 1e-3 |
| Learning scheduler | cos_with_min_lr | cos_with_min_lr | cos_with_min_lr |
| Min LR | 1e-5 | 1e-5 | 1e-5 |
| Warmup ratio | 0.015 | 0.015 | 0.015 |
| Adam $\beta_1$ | 0.9 | 0.9 | 0.9 |
| Adam $\beta_2$ | 0.95 | 0.95 | 0.95 |
| Weight decay | 1e-1 | 1e-1 | 1e-1 |
| Total batch size | 64 | 64 | 64 |
| Segment length (tokens) | 4096 | 4096 | 4096 |
| Training steps | 245K | 245K | 409K |

# D  TRAINING CONFIGURATIONS

Table 4: Model and training configurations across scales and variants.

| | Baseline | | | MH-FFN | | PKV | FlashMHF | | |
|---|---|---|---|---|---|---|---|---|---|
| | 128M | 370M | 1.3B | 128M | 370M | 370M | 128M | 370M | 1.3B |
| Params (M) | 117.96 M | 388.05 M | 1.323 B | 115.70 M | 379.79 M | 386.55 M | 115.77 M | 390.08 M | 1.321 B |
| $n_{\text{layers}}$ | 12 | 24 | 24 | 10 | 20 | 24 | 10 | 21 | 20 |
| $d_{\text{model}}$ | 768 | 1024 | 2048 | 768 | 1024 | 1024 | 768 | 1024 | 2048 |
| $n_{\text{att-heads}}/d_{\text{head}}$ | 12/64 | 16/64 | 32/64 | 12/64 | 16/64 | 16/64 | 12/64 | 16/64 | 32/64 |
| Intermediate size | 2048 | 2752 | 5504 | 2048 | 2752 | 3072 | 2048 | 2688 | 5760 |
| $n_{\text{ffn-heads}}(H)/d_h$ | -/- | -/- | -/- | 6/128 | 8/128 | 8/128 | 6/128 | 8/128 | 16/128 |
| Sub-network Num $E$ | - | - | - | - | - | - | 8 | 7 | 15 |
| Training steps | 245K | 245K | 409K | 245K | 245K | 245K | 245K | 245K | 409K |
| Learning rate | 3e-3 | 1.5e-3 | 1e-3 | 3e-3 | 1.5e-3 | 1.5e-3 | 3e-3 | 1.5e-3 | 1e-3 |
| Training tokens | 60B | 60B | 100B | 60B | 60B | 60B | 60B | 60B | 100B |
| Hidden act | silu | silu | silu | silu | silu | silu | silu | silu | silu |
| initializer_range | 0.02 | 0.02 | 0.02 | 0.02 | 0.02 | 0.02 | 0.02 | 0.02 | 0.02 |
| max-position_embeddings | 4096 | 4096 | 4096 | 4096 | 4096 | 4096 | 4096 | 4096 | 4096 |
| pretraining_tp | 1 | 1 | 1 | 1 | 1 | 1 | 1 | 1 | 1 |
| rms_norm_eps | 1e-05 | 1e-05 | 1e-05 | 1e-05 | 1e-05 | 1e-05 | 1e-05 | 1e-05 | 1e-05 |
| rope_scaling | null | null | null | null | null | null | null | null | null |
| tie_word_embeddings | true | false | false | true | false | false | true | false | false |
| torch_dtype | bfloat16 | bfloat16 | bfloat16 | bfloat16 | bfloat16 | bfloat16 | bfloat16 | bfloat16 | bfloat16 |
| vocab_size | 50432 | 50432 | 50432 | 50432 | 50432 | 50432 | 50432 | 50432 | 50432 |

# E   EFFICIENCY BENCHMARK CONFIGURATION DETAILS ON HOPPER

Table 5: FlashMHF vs. SwiGLU vs. MH-FFN across sequence lengths on **Hopper**.

| Arch | bs | $L$ | $d_e$ | $H$ | $E$ | $d_h$ | Latency (ms) | | | Memory (MB) | | |
|------|----|-----|-------|-----|-----|-------|-----------|--------|--------|-----------|--------|--------|
| | | | | | | | FlashMHF | SwiGLU | MH-FFN | FlashMHF | SwiGLU | MH-FFN |
| Hopper | 8 | 192 | 384 | 16 | 22 | 128 | 6.80 | 6.24 | 101.40 | 184.10 | 251.00 | 2702.10 |
| Hopper | 8 | 384 | 384 | 16 | 22 | 128 | 13.20 | 12.24 | 146.60 | 218.20 | 370.00 | 4462.00 |
| Hopper | 8 | 768 | 384 | 16 | 22 | 128 | 24.40 | 24.72 | 235.60 | 286.50 | 606.00 | 7982.20 |
| Hopper | 8 | 1536 | 384 | 16 | 22 | 128 | 48.60 | 48.96 | 401.60 | 423.00 | 1070.00 | 15021.50 |
| Hopper | 8 | 1920 | 384 | 16 | 22 | 128 | 59.60 | 63.12 | 484.60 | 491.20 | 1306.00 | 18541.60 |
| Hopper | 8 | 2880 | 384 | 16 | 22 | 128 | 90.40 | 94.56 | 688.60 | 661.90 | 1892.00 | 27341.90 |
| Hopper | 8 | 4032 | 384 | 16 | 22 | 128 | 126.40 | 127.44 | 933.20 | 866.30 | 2592.00 | 37902.30 |
| Hopper | 8 | 8064 | 384 | 16 | 22 | 128 | 254.60 | 267.60 | 1793.40 | 1582.20 | 5050.00 | 74864.00 |
| Hopper | 8 | 16128 | 384 | 16 | 22 | 128 | 497.40 | 535.20 | OOM | 3016.20 | 9966.00 | OOM |

# F   LLM USAGE

We acknowledge the use of a Large Language Model (LLM), specifically Google's Gemini and OpenAI's ChatGPT, as a writing assistant during the preparation of this manuscript. The LLM's role was strictly limited to improving the clarity, conciseness, and grammatical correctness of the text. All scientific contributions presented in this paper—including the formulation of the core ideas, the design of the FlashMHF architecture, the experimental methodology, and the analysis and interpretation of results—are the original work of the human authors.

