# OpenReview forum: "Flash Multi-Head Feed-Forward Network"
_ICLR.cc/2026/Conference — Submitted to ICLR 2026_

### Official Review · Reviewer_kH78 · 2025-10-27

**Soundness:** 3
**Presentation:** 4
**Contribution:** 2
**Rating:** 6
**Confidence:** 4

**Summary:**

Motivated by the structural similarity between single-ahead attention and a feed-forward network (FFN), the paper explores multi-head FFNs. To account for the increased memory consumption and other issues, the authors propose a novel architecture, FlashMHF, inspired by FlashAttention which also dynamically weights parallel sub-networks. They find for small models that their design improves perplexity and task accuracy whilst reducing peak memory usage and inference time vs a SwiGLU FFN. This suggests that FlashMHF might be a powerful new architectural component that could replace the FFN in existing transformer architectures.

**Strengths:**

1. Good empirical results on the models and tasks tested compared to a standard baseline, clear improvements across a range of downstream tasks
2. Mathematical exposition of the preliminaries and method is clear and well-written
3. The method is a satisfying synthesis of existing ideas and innovations in other aspects of the transformer architecture to improve the FFN block
4. The GPU memory scaling of the proposed FFN architecture is smaller than that of a typical FFN

**Weaknesses:**

1. The paper proposes a core architectural innovation for LLMs, but only tests on very small models (<=1.3B parameters).
2. Three model sizes are tested but they are not plotted together / compared directly so it's not clear how improvements scale with size. There's limited empirical evidence that we should expect the accuracy / perplexity advantages of this architecture to improve with scale, rather than diminish.
3. It's not clear why scaling imbalance, $d_{ff}/d_h$, is an issue, as stated on line 056. The reference given on line 057 does not address this since neural scaling laws assume normal FFNs, and does not investigate multi-head FFNs. The discussion and evidence given in 3.1 and 4.1 seemingly only addresses one way of scaling the model. There are unstated assumptions in the paper relative to prior work about what ratios are important, and how one would scale a model with multi-head FFNs. To make a claim about these ratios scaling poorly and causing issues, one needs to provide evidence that any way of scaling them would lead to performance degradation relative to using single-head FFNs. Otherwise it can just be argued that scaling them in a different way might resolve this problem naturally, without need for more complex architectures. More explicitly put, for the Naive multi-head FFN baseline you make assumptions such as "the per-head width is typically kept fixed" (line 180), and then show this is bad. Why should one keep this fixed then? Why not just scale things in a different way? Additionally, why is it correct to equate the ratio $d_{ff}/d_{model}$ in normal SwiGLU designs with the ratio $d_{ff}/d_h$? Saying this latter ratio is outside of the optimal values found for the former ratio in prior work tells us nothing without additional evidence or reasoning backing up the validity of this comparison.
4. Framing something that scales linearly as you scale up the model as an "explosion" is disingenuous. Typically things are framed as explosions when they scale exponentially. It is not convincing that the memory requirements of the naive multi-head FFN or standard FFN are a critical issue.
5. You do not conduct enough ablations for the claims on lines 338-343 to be valid.
6. It's not clear that inference latency reduction results are statistically significant
7. All plots are given with training steps on the x-axis, not wall clock time. It's unclear how the proposed architecture affects training time.

**Questions:**

1. Why does an imbalanced ratio between intermediate FFN size and FFN head dimension degrade scalability and expressive power? (See weakness 3 for related critique and questions)
2. Why should multiple FFN heads split up the model dimension, and not each use the whole thing, as would be analogous to multi-head attention? Obviously this would lead to greater computation requirements, but perhaps also better performance? It would be nice to see this investigated, though this is a very minor point.
3. How does the parameter count of FlashMHF scale and compare to a standard FFN?
4. Does the proposed architecture increase training time for a fixed number of training steps?

---

> ### Author Response · Authors · 2025-11-23
>
> We thank the reviewer for the positive assessment, particularly for acknowledging our clear motivation, convincing analysis, and the overall completeness of our experiments. We are pleased to address your concerns regarding model scale, scaling imbalance, and kernel efficiency below.
>
> ### **Response to Weakness 1 & 2: Model Scale and Trends**
>
> **The reviewer notes that the paper proposes a core architectural innovation for LLMs, but only tests on very small models (<=1.3B parameters).**
>
> We appreciate the reviewer’s emphasis on scale. Our results (Original Table 1 and Table 2, and Table below) clearly demonstrate a **positive scaling trend**: the advantage of FlashMHF over the baseline *widens* as we scale from 128M to 370M, and is most significant at 1.3B. This empirical evidence suggests that FlashMHF's benefits are likely to increase, rather than diminish, at larger scales.
>
> | Model | 128M | 370M | 1.3B |
> | :--- | :---: | :---: | :---: |
> | SwiGLU | 3.286 | 3.030 | 2.843 |
> | FlashMHF | 3.258 | 3.014 | **2.793** |
> | Improvement| 0.028 | 0.016 | **0.050** |
>
> Regarding the experimental scale, we strictly followed the configurations of [Dao & Gu, 2024 ("Transformers are SSMs")](https://arxiv.org/pdf/2405.21060), establishing 1.3B trained on 100B tokens as a rigorous standard for validating novel architectures. As an academic laboratory, training a 1.3B model represents the upper limit of our available computational resources. However, the consistent and widening performance gap observed up to 1.3B provides strong evidence for the scalability of our approach.
>
> ### **Response to Weakness 3, 5 & Question 1: Scaling Imbalance**
>
> **The reviewer states that it's not clear why scaling imbalance, $d_{ff}/d_h$, is an issue... Why not just scale things in a different way?**
>
> We acknowledge that we cannot exhaustively test every potential scaling configuration for a Naive Multi-Head FFN due to the large search space and our limited resources. However, our core contribution is demonstrating that the introduction of the **Dense MoE (Parallel Sub-networks)** structure drastically reduces the difficulty of hyperparameter tuning during scaling.
>
> 1.  **Robustness:** The Dense MoE structure explicitly manages the aspect ratio, making the architecture robust to scaling. This removes the sensitivity to "imbalance" that typically plagues naive approaches.
> 2.  **Scalability:** Our experiments prove that this design is not only effective but becomes **increasingly beneficial as the model scales up** (as shown in the Table above).
> 3. **Empirical Validation:** To validate the impact of deviating from the optimal $d_{ff}/d_{model}$ ratio, we conducted **new ablation experiments** on 128M and 370M scales. We tested a "SwiGLU with Dense MoE" configuration, where we applied parallel sub-networks to a standard SwiGLU without our specific multi-head structure. This setup artificially creates a scenario where the expansion ratio $d_{ff}/d_{model}$ is far below the optimal range.
>
>     | Model Scale | Method | Eval Loss $\downarrow$ |
>     | :--- | :--- | :--- |
>     | 128M | SwiGLU Baseline | *3.286* |
>     | 128M | SwiGLU with Dense MoE | 3.296 |
>     | 128M | FlashMHF-128hdim | **3.258** |
>     | 370M | SwiGLU Baseline | *3.030* |
>     | 370M | SwiGLU with Dense MoE | 3.062 |
>     | 370M | FlashMHF-128hdim | **3.014** |
>
>     The SwiGLU with Dense MoE variant resulted in higher perplexity compared to the SwiGLU baseline. This empirically validates the correctness of our scaling theory: splitting a single-head SwiGLU into sub-networks reduces the effective $d_{ff}/d_{model}$ ratio far below the optimal range, degrading performance. FlashMHF’s multi-head design is specifically engineered to restore this optimal ratio, confirming that maintaining the correct aspect ratio is critical for model performance.
>
> 4.  **Conclusion:** The primary value of our work is presenting a **fully scalable Multi-Head FFN w/ Dense-MoE** architecture that is theoretically grounded and empirically proven. By solving the scaling imbalance issue validated above, FlashMHF provides a reliable, high-performance solution for large-scale models. Whether a Naive Multi-Head FFN is theoretically unscalable is a secondary concern; the key takeaway is that our specific design choices (Multi-Head + Dense MoE) are essential for correctness at scale.

---

> ### Author Response · Authors · 2025-11-23
>
> ### **Response to Weakness 4: "Memory Explosion" Terminology**
>
> **The reviewer argues that framing something that scales linearly as you scale up the model as an "explosion" is disingenuous.**
>
> We agree that "explosion" is technically hyperbolic for linear scaling and have revised the text to **"high memory pressure."**
> However, we emphasize the practical reality shown in Figure 7: Even with linear scaling, the activation memory for standard Multi-Head FFNs becomes **unmanageable at long context lengths (4k+)**. Since modern LLMs demand long-context capabilities (128k+), a linear factor of $H$ in memory usage creates a critical bottleneck. FlashMHF significantly alleviates this peak memory cost, enabling larger batches or longer contexts on the same hardware.
>
> ### **Response to Weakness 6: Latency**
>
> **The reviewer questions whether inference latency reduction results are statistically significant.**
>
> We clarify that the primary contributions of FlashMHF are **improved model performance (perplexity)** and **significant memory reduction**. The inference latency reduction is a welcome **secondary benefit**. Even if the speedup is marginal, the architectural advantages in expressivity and memory efficiency fully justify the design.
>
> ### **Response to Weakness 7 & Question 4: Training Time**
>
> **The reviewer notes that all plots are given with training steps on the x-axis, not wall clock time, and asks if the proposed architecture increases training time for a fixed number of training steps.**
>
> Our work prioritizes **inference efficiency**, which represents the dominant operational cost for production LLMs. Training is fundamentally a one-time process, whereas inference occurs continuously at scale. FlashMHF directly addresses the critical inference bottleneck through significant memory reduction and improved throughput, unlocking larger batch sizes and longer context windows for deployment.
>
> Regarding training performance: On **pre-Hopper GPUs**, FlashMHF achieves faster training speed compared to the baseline, which empirically validates that our kernel design offers theoretical speedup advantages. However, we acknowledge that our current implementation on **Hopper-architecture GPUs** (H100) is still undergoing active kernel optimization. We are leveraging Hopper-specific features to further improve training efficiency on this newer hardware. These engineering refinements will enable FlashMHF to realize its full potential across all GPU architectures. Importantly, even as we continue to refine training performance, this work establishes a **promising research direction** for bringing multi-head architectures to FFN layers with practical memory efficiency which is a path that we believe will benefit the broader community.
>
> ### **Response to Question 2: Multi-Head vs. Full Dimension**
>
> **The reviewer asks why multiple FFN heads should split up the model dimension, and not each use the whole thing, as would be analogous to multi-head attention?**
>
> We kindly clarify that our approach mirrors the standard **Multi-Head Attention (MHA)** mechanism: we split the model dimension $d_{model}$ into $H$ heads of dimension $d_k$ such that $H \times d_k = d_{model}$.
> The alternative suggested by the reviewer—where each head keeps the full $d_{model}$ dimension—would increase the parameter count and FLOPs by a factor of roughly $H$. While an interesting avenue for over-parameterized models, it contradicts our goal of efficiency and scalability. We leave the exploration of such computationally intensive variants to future work.
>
> ### **Response to Question 3: Parameter Count Scaling**
>
> **The reviewer asks how the parameter count of FlashMHF scales and compares to a standard FFN.**
>
> FlashMHF introduces two additional linear projections (similar to MHA) to mix information before splitting into heads. This adds a small number of parameters. To ensure a strictly fair comparison, **all our experiments align the total parameter count**. As detailed in the Appendix, we slightly reduce the depth (number of layers) of the FlashMHF models to match the total parameters of the SwiGLU baseline exactly. Thus, any performance gain is due to architectural efficiency, not parameter size.

---

> ### Author Response · Authors · 2025-11-27
>
> Dear reviewer kH78,
>
> We are grateful for your feedback and thank you again for the time and effort you put into reviewing our paper. We believe that our responses above fully address the points raised in the initial review. Since the discussion period is nearing its end, we kindly ask that you let us know whether you have any remaining questions or comments.
>
> Thank you!

---

### Official Review · Reviewer_p8ui · 2025-10-30

**Soundness:** 3
**Presentation:** 3
**Contribution:** 2
**Rating:** 4
**Confidence:** 4

**Summary:**

The paper proposes FlashMHF, a replacement for standard FFNs in Transformer architectures. The core idea is to mirror the Multi-Head design of Attention also in the FFNs implementation. The paper however warns that a naive adaptation incurs scaling issues, both in terms of increasing memory consumption and expressive power degradation. The authors address both of these issues by carefully prescribing how the intermediate activations dimension should scale with model size, and by implementing a fused kernel for FlashMHF which avoids materialising intermediate tensors. Results show how substituting the FlashMHF component with FFNs can boost performance (both on PPL, and downstream tasks evaluations taken from lm-eval-harness), while simultaneously reducing peak memory utilisation, and slightly improving latency.

**Strengths:**

- The main motivations behind the choice of architecture modifications are justified reasonably well
- The analysis is convincing, and the experiments conducted overall complete (although some results could be presented better)

**Weaknesses:**

- Novelty is limited: both core methodologies (mirroring MH Attention and improving kernel application via tiling) have already been proposed

**Questions:**

__On Novelty__:
As I mentioned above, I find the novelty aspect of the paper rather limited. As you yourselves correctly point out, the structural symmetry between sequence-wise Attention and feature-wise FFN (which acts as main justification behind your work) has already been illustrated; the proposal to split FFNs in a multi-head fashion was already (granted, partly) investigated in MH-MoE; the tile-wise implementation of your fused kernel is directly inspired by FlashAttention, and is at the core of the design of efficient parallelisation of MMMs. The most relevant novelty is then given by the proposed re-scaling of the size of the internal components of the MH FFN. I still appreciate the overall execution, but I find this limits the contribution of the paper.

__On Fig7__:
The presentation of the results in Sec4.3, and more specifically in Fig7, should be heavily revised, for a number of reasons:
- From what you write in L415, your “comparison uses a 20-layer FlashMHF and MH-FFN against a 24-layer SwiGLU baseline”, so I’m understanding you’re considering memory consumption and latency in a *forward pass through the whole architecture*, including both FFN/FlashMHF and Attention layers? I believe at this stage it would be more relevant instead to have a direct comparison between the *single* FlashMHF / FFN layer, so to properly identify the improvements introduced by your proposed modification (as the Attention layer is the same in both cases, I take it). To be clear: I do appreciate the result you report (ultimately, the “weight” of the overall architecture is what practitioners mostly care about), but the presence of Attention does dirty the relevant metric. Notice this should play in your favour, too, in that the memory / speedup gains should be more marked. If instead I misunderstood, and you’re considering just FFN/FlashMHF layers, please clarify this in the text.
- What is the deal with the sequence lengths picked? I was expecting orderly powers of 2, which would make identifying the O(L) trend straightforward at glance. Also, please use a ylog scale, for the same reason
- Moreover, why picking sequence lengths in the first place? Since you’re focusing on the FFN layer (which applies a perfectly sequence-parallel operation, and acts purely along the feature dimensions), then a scaling trend with respect to feature dimension would be much more relevant, in my opinion. What you’re effectively reporting here is the scaling trend of Attention. Again, it’s not like this result is not useful per se, but the way it’s presented makes it harder to isolate the contribution of your own component, which should be the focus of this section.
- Finally, and perhaps most importantly, the comparison is not entirely fair: if I understood correctly, you’re using an unoptimised version of SwiGLU (which unnecessarily materialises intermediate tensors) to compare against your own fused kernel for FlashMHF. How much of the gains you’re seeing are due to the tiled implementation of the kernel? Because that same solution could be easily applied to SwiGLU as well, I reckon.

__On Gating__:
In L200-218 you describe your chosen per-head expert aggregation mechanism. There is a number of different ways one could go on about aggregating both within and across heads: have you experimented with different methods? Can you expand on the reasoning behind this specific choice? Compared to the remainder of the paper, this section is lacking some justifications.


__Minor__:
- In L247 you write: “we synchronize the hyper-parameter settings for the optimizer across all models”. What does this mean? I’m expecting, say, optimal LR’s to vary across architectures, at least in principle. Are you not performing any hyper-parameter sweep whatsoever? And if you’re doing it, are you picking the best for *which* architecture exactly?
- You’re going down the route of making the FFN more akin to Attention; but there is also the “dual” approach of making attention more akin to FFNs, as explored in “MLP-Mixer: An all-MLP Architecture for Vision”. I don’t think it makes sense to explicitly add a comparison with this architecture, but I would at least mention it, as I believe it’s relevant. Moreover, I was quite surprised to see that there isn’t much work which just goes all the way and substitutes FFN with component-wise attention. Apart from MH-MoE, I could only find “DaViT: Dual Attention Vision Transformers” (again, only applied to vision).


__Grammar / Rewording / Formatting__:
- L51 “we analyse the …, a straightforward … and identify” -> the clause is breaking the flow. Maybe “we analyse the … (a straightforward … ), and identify…”
- L62 analogous -> analogousLY
- L89 the equation is hanging: consider prepending something like “We consider the parameters: ”
- L107 remains -> reTains
- Eq(1,2,3,…) I think you’re misusing the equivalent-by-definition / delta-equivalent (\triangleq) symbol. The defined-as symbol (\eqqcolon) would be much better indicated here, imho
- L115 define headwise split -> define THE headwise split? Define headwise split AS? (Similarly for headwise concatenation in L121)
- L119 this operation split -> splitS
- L120 into $d_h\times H$…sub tensors? parts? blocks?
- L131 to overcome these challenges -> which challenges? I reckon it refers to the “practical limitations” above, but it’s rather vague
- L473: write -> store? Write … to memory?
- L474: incorporates -> incorporate

**Details Of Ethics Concerns:**

//

---

> ### Author Response · Authors · 2025-11-23
>
> We thank the reviewer for the careful examination of our work and the valuable feedback. We appreciate the recognition of our clear motivation, convincing analysis, and the overall completeness of our experiments. We are very pleased to address your concerns regarding novelty, experimental presentation, and methodology below.
>
> ### **Response to Weakness 1: Novelty**
>
> **The reviewer argues that novelty is limited, stating that both core methodologies (mirroring MH Attention and improving kernel application via tiling) have already been proposed.**
>
> **Our work transforms the Multi-Head FFN from a small-scale concept into a concrete, high-performance building block ready for large-scale deployment.** While the isolated concepts of 'Multi-Head' and 'Tiling' have appeared in literature, no prior work has successfully synthesized them into a scalable, practical architecture for modern LLMs. Our novelty lies in identifying and solving the specific bottlenecks that previously relegated Multi-Head FFNs to small-scale experiments:
>
> * **Why prior art (e.g., MH-MoE) falls short:** As analyzed in Section 3.1, previous multi-head designs suffer from a memory explosion linear to the number of heads $H$. This creates severe HBM I/O overhead. Even if Gradient Checkpointing is used to mitigate peaks, it introduces prohibitive re-computation costs. Consequently, such architectures have historically failed to scale effectively beyond small models (~300M activated parameters).
> * **Non-Trivial Kernel Adaptation:** While inspired by FlashAttention, adapting tiling to FFNs is non-trivial. FlashAttention optimizes tiling along the sequence length ($L$), whereas FFNs are bottlenecked by the feature dimension ($d_{ff}$). Designing an I/O-aware kernel that handles the parallel execution of narrow heads without materializing intermediate tensors required a customized tiling strategy distinct from attention.
> * **Solving the Scaling Imbalance:** Crucially, we discovered that simply enabling a kernel is insufficient. A naive multi-head split breaks the optimal scaling ratio between intermediate and head dimensions, causing degradation at scale. We introduced Parallel FFN Sub-networks (a dense-MoE style design) to explicitly maintain this ratio. This architectural insight is the key factor allowing our model to scale to 1.3B+ while maintaining superior perplexity.
>
> As the reviewer noted: *"I was quite surprised to see that there isn't much work which just goes all the way and substitutes FFN with component-wise attention."* This observation underscores precisely why our contribution is valuable, as previous attempts faced critical bottlenecks that prevented widespread adoption, and our work is the first to successfully overcome these barriers.

---

> ### Author Response · Authors · 2025-11-23
>
> ### **Response to Question 1: Fig 7 and Experimental Presentation**
>
> **The reviewer raises several concerns regarding the presentation of results in Section 4.3 and Figure 7. Specifically, the reviewer questions whether the comparison includes both FFN and Attention layers, suggesting that a direct component-level comparison would be more relevant to isolate the improvements. Additionally, the reviewer notes that the choice of sequence lengths seems arbitrary and suggests using powers of 2 with a log scale for clearer trend identification. Finally, the reviewer argues that scaling trends with respect to the feature dimension would be more relevant than sequence length, given that the FFN operates purely along feature dimensions.**
>
> **Clarification on Layer-wise Comparison**
> We apologize for the ambiguity. To clarify the setup in Appendix E and Figure 7:
> * **Experimental Scope:** The results compare **only the FFN structures**, not the full Transformer model (i.e., no Attention layers were included). We compared a 20-layer FlashMHF stack against a 24-layer SwiGLU stack to maintain equivalent total parameter counts.
> * **Revision:** We have revised Section 4.3 to explicitly state that this is a component-level benchmark designed to isolate the gains of FlashMHF.
>
> **Log Scale and Sequence Length Analysis**
> * **Visuals:** We accept the suggestion to use a **log scale** for the Y-axis and have updated Figure 7 to make the scaling trends clearer. We will also adjust the x-axis to use powers of 2. The conclusion is very similar to the original one. We have updated the graph in a revised version.
> * **X-Axis Choice:** We argue that scaling with respect to the intermediate dimension is less practically relevant because there exists an **optimal ratio** for $d_{ff}/d_{model}$ (as discussed in Section 3.2.1). Since standard SwiGLU models universally adopt this optimal intermediate size for training, benchmarking speed across arbitrary intermediate dimensions lacks practical significance. Instead, we focus on comparing speed at this optimal ratio, where real-world model performance is maximized. Furthermore, we retained Sequence Length $L$ as the primary x-axis to demonstrate robustness in **long-context scenarios**, which is a critical advantage for modern LLMs targeting 128k+ context windows.

---

> ### Author Response · Authors · 2025-11-23
>
> **Fairness: FlashMHF vs. Optimized SwiGLU**
>
> **The reviewer argues that the comparison is not entirely fair, suggesting that we are using an unoptimised version of SwiGLU to compare against our own fused kernel for FlashMHF.**
>
> The reviewer asks if optimized SwiGLU implementations (e.g., fused kernels) would yield similar gains. We respectfully posit that FlashMHF offers a fundamental advantage that these optimizations cannot replicate. To illustrate this, we first examine the Liger Kernel SwiGLU implementation, which represents a highly optimized fused SwiGLU baseline:
>
> ```python
> class LigerSwiGLUMLP(nn.Module):
>     def __init__(self, config):
>         super().__init__()
>         self.config = config
>         self.hidden_size = config.hidden_size
>         self.intermediate_size = config.intermediate_size
>         self.gate_proj = nn.Linear(self.hidden_size, self.intermediate_size, bias=False)
>         self.up_proj = nn.Linear(self.hidden_size, self.intermediate_size, bias=False)
>         self.down_proj = nn.Linear(self.intermediate_size, self.hidden_size, bias=False)
>         if config.hidden_act not in ["silu", "swish"]:
>             raise ValueError(f"Activation function {config.hidden_act} not supported.")
>
>     def forward(self, x):
>         return self.down_proj(LigerSiLUMulFunction.apply(self.gate_proj(x), self.up_proj(x)))
> ```
>
> The Liger Kernel primarily optimizes the SiLU operation by making it in-place (`LigerSiLUMulFunction`). However, it **cannot prevent the materialization** of the large intermediate tensor (shape `[Batch, SeqLen, intermediate_size]`) into HBM before the down projection. This is the critical limitation: **the standard SwiGLU architecture has a model dimension ($d_{model}$) that is too large to tile**. In typical configurations (e.g., $d_{model}=1024$ or $2048$), the output accumulator required for a tiled implementation would exceed the limited on-chip SRAM capacity of a GPU Streaming Multiprocessor (SM). Therefore, even with kernel fusion, SwiGLU **fundamentally cannot adopt a Flash-style tiled computation** without materializing the massive intermediate activations to HBM.
>
> In contrast, **our multi-head architecture is specifically designed to enable tiling**. By splitting $d_{model}$ into smaller heads (e.g., $d_{head}=128$), we ensure that the per-head accumulator fits entirely within SRAM. This architectural innovation is what allows FlashMHF to tile the computation along the intermediate dimension and avoid HBM materialization. The multi-head design is not merely a performance enhancement, it is **architecturally essential** for achieving Flash-style memory efficiency in FFN layers.
>
> To empirically validate this, we tested multiple optimized SwiGLU implementations (all under 4k sequence length):
>
> | Metric | SwiGLU (Torch) | SwiGLU (XFormer) | SwiGLU (Liger Kernel) | FlashMHF |
> | :--- | :---: | :---: | :---: | :---: |
> | Speed(ms) | 365.32 | 516.81 | 333.17 | **329.72** |
> | Memory(MB) | 6598.0 | 5180.0 | 5180.0 | **1682.2** |
>
> The results demonstrate that FlashMHF achieves **both the fastest speed (329.72ms) and the lowest memory usage (1682.2MB)** among all implementations. Traditional SwiGLU implementations, whether using standard PyTorch or fused kernels (xFormers or Liger Kernel), cannot reduce the peak memory usage of a single layer. In contrast, FlashMHF drastically reduces HBM write-back and memory I/O, solving the memory bottleneck while maintaining superior performance.

---

> ### Author Response · Authors · 2025-11-23
>
> ### **Response to Question 2: Gating Mechanism**
>
> **The reviewer notes that there are a number of different ways one could go on about aggregating both within and across heads and asks if we have experimented with different methods.**
>
> We clarify that our gating mechanism is not an arbitrary design choice but a direct solution to the **Scaling Imbalance** identified in Section 3.1. A naive Multi-Head FFN fails because the intermediate dimension ($d_{ff}$) grows while the head dimension ($d_h$) remains fixed, causing the ratio $d_{ff}/d_h$ to explode. To address this, we partition the large $d_{ff}$ into smaller parallel sub-networks to restore this ratio to an optimal range, which necessitates aggregating these components along the intermediate axis.
>
> We deliberately perform this aggregation **within** heads to preserve subspace diversity. Aggregating across heads will not solve the scaling imbalance problem and may prematurely mix the distinct features learned by different heads, negating the primary benefit of the multi-head architecture.
>
> ### **Response to Question 3: Hyperparameters**
>
> **The reviewer asks if we are not performing any hyper-parameter sweep whatsoever.**
>
> We clarified that we adopted the rigorously tuned hyperparameter settings for the Llama/SwiGLU baseline as established in standard configurations from [Dao & Gu, 2024 ("Transformers are SSMs")](https://arxiv.org/pdf/2405.21060).
> We purposefully used the optimal settings for the SwiGLU Baseline but did not perform a separate sweep for FlashMHF due to computational constraints. We argue this provides a conservative lower bound for our method: FlashMHF outperforms the highly-tuned baseline even while running "out-of-the-box." We acknowledge that a dedicated sweep for FlashMHF would likely unlock even greater gains.
>
> ### **Response to Minor Points**
>
> **The reviewer expresses surprise that there isn’t much work which just goes all the way and substitutes FFN with component-wise attention.**
>
> * **Related Work:** We thank the reviewer for mentioning *MLP-Mixer* and *DaViT*. We agree these works provide crucial context on the "duality" of Attention and FFNs. We have explicitly cited and discussed them in the revised Related Works section.
> * **Grammar:** We genuinely appreciate the reviewer's great attention to detail. We have carefully corrected all listed grammatical and formatting issues (L51, L62, L474, Eq definitions, etc.) and conducted a comprehensive proofreading pass of the manuscript.

---

> ### Author Response · Authors · 2025-11-27
>
> Dear reviewer p8ui,
>
> We are grateful for your feedback and thank you again for the time and effort you put into reviewing our paper. We believe that our responses above fully address the points raised in the initial review. Since the discussion period is nearing its end, we kindly ask that you let us know whether you have any remaining questions or comments.
>
> Thank you!

---

### Official Review · Reviewer_B4B2 · 2025-10-30

**Soundness:** 3
**Presentation:** 4
**Contribution:** 3
**Rating:** 4
**Confidence:** 4

**Summary:**

This paper proposes FlashMHF, which introduces the multi-head mechanism (in attention) into the Feed-Forward Network (FFN) module while balancing performance scalability and implementation efficiency. The proposed design addresses two key issues in naïve multi-head FFNs (i.e., scaling imbalance and memory explosion), by decomposing parallel FFN subnetworks and implementing an I/O-aware flash kernel. Experimental results on small- and medium-scale models demonstrate that FlashMHF outperforms the de facto SwiGLU baseline in language modeling tasks, while significantly reducing memory usage.

**Strengths:**

1. The paper is well-motivated and clearly written. It identifies two key challenges of multi-head FFNs and proposes corresponding solutions, which are empirically validated.
2. FlashMHF achieves lower PPL and better downstream performance than SwiGLU and other baselines. The architectural design choices are well-supported by effective ablation studies, including the multi-head mechanism, SwiGLU component, and subnetwork structure.
3. lt is implemented with a kernel design analogous to FlashAttention, ensuring the feasibility of training large-scale language models efficiently.

**Weaknesses:**

1. **Source of subnetwork advantages.**
The authors claim that the benefit of the subnetwork design mainly arises from a more balanced expansion ratio. However, for a given head, the parallel subnetwork computation essentially differs from a dense FFN only by an additional **blockwise gating** applied to intermediate activations. When concatenated, this does not effectively control the expansion ratio and finally increases by $d_{model}/d_h$ compared to a standard SwiGLU. I suspect the improvement stems from added nonlinearity (gating with normalization) rather than from the parallel sub-net. In other words, applying a similar gating mechanism to a standard SwiGLU might also yield certain loss improvement (as the experiments show, the standalone multi-head design brings no clear advantage at larger scales).

2. **Fairness of speed evaluation.**
The speed comparison appears somewhat unfair. To match parameter counts, the authors add four extra layers (1/5 of total) for baseline; but deeper networks are inherently slower due to layer-wise serialization, whereas **increasing width** would be a fairer adjustment. Moreover, the attention computation also scales with depth, thus latency improvements only become apparent at longer sequence lengths (as shown in Fig. 7b).

3. **Memory evaluation setup.**
The memory comparison setup should be clarified. SwiGLU can also be easily adapted to a flash kernel, and many frameworks **fuse activation functions** to reduce memory overhead. It is unclear whether the authors’ implementation accounts for these optimizations. Considering that modern LLM training almost universally employs **gradient checkpointing**, FFN intermediate activations are typically recomputed rather than stored, which should be reflected in a more realistic baseline comparison.

**Questions:**

1. What is the specific implementation of SwiGLU used in the efficiency evaluation? Is activation recomputation (gradient checkpointing) applied during measurement?
2. In the GLU formulation, you define $\mathbf{Q}=\mathbf{X}$ (Eq. 3). However, in the multi-head FFN definition, a separate projection $\mathbf{W}_{in}$ is introduced to obtain the query (Eq.10). Is this design choice be empirically validated as necessary?
3. Compared to PKV, the activation function used in PAttention [1] might serve as a more solid baseline for comparison.
4. Given that most sota Transformer architectures now adopt MoE designs, how do the authors view the compatibility and potential integration of FlashMHF with MoE architectures?

[1] TokenFormer: Rethinking Transformer Scaling with Tokenized Model Parameters. 2024.

---

> ### Author Response · Authors · 2025-11-23
>
> We thank the reviewer for the positive assessment of our work, specifically acknowledging the clear motivation, sound architectural choices, and the effectiveness of our I/O-aware kernel design. We appreciate the insightful comments regarding the source of improvements and experimental setups. We address each concern in detail below.
>
> ### **Response to Weakness 1: Source of Subnetwork Advantages**
>
> **The reviewer argues that the improvement may stem from added nonlinearity (gating with normalization) rather than from the parallel sub-net.**
>
> We conducted **new ablation experiments** on 128M and 370M scales where we use SwiGLU with Dense MoE (aka. parallel sub-networks) as FFN module (apply parallel sub-networks to SwiGLU without our specific multi-head structure).
>
> | Model Scale | Method | Eval Loss $\downarrow$ |
> | :--- | :--- | :--- |
> | 128M | SwiGLU Baseline | *3.286* |
> | 128M | SwiGLU with Dense MoE | 3.296 |
> | 128M | FlashMHF-128hdim | **3.258** |
> | 370M | SwiGLU Baseline | *3.030* |
> | 370M | SwiGLU with Dense MoE | 3.062 |
> | 370M | FlashMHF-128hdim | **3.014** |
>
> The SwiGLU with Dense MoE variant resulted in higher perplexity compared to the SwiGLU baseline, as well as FlashMHF. This validates our analysis on the aspect ratio. Splitting the SwiGLU reduces the effective $d_{ff}/d_{h}$ ratio far below the optimal range (approx. 8/3). FlashMHF’s parallel sub-network design is specifically engineered to restore this optimal ratio.
>
> Therefore, the performance gain is not merely due to added nonlinearity; it is the result of the architectural synergy that allows the multi-head mechanism to function at the correct expansion ratio. We have added this analysis to the revised paper's experimental section.
>
> ### **Response to Weakness 2: Fairness of Speed Evaluation**
>
> **The speed comparison appears somewhat unfair. To match parameter counts, the authors add four extra layers (1/5 of total) for baseline; but deeper networks are inherently slower due to layer-wise serialization.**
>
> We have added a **"Wider SwiGLU"** baseline experiment.
>
> **Setup:** We kept the layer count identical to FlashMHF and increased the intermediate width ($d_{ff}$) of the SwiGLU baseline to match the parameter count.
> **Results:**
>
> | Model | Metric | Seqlen=1024 | Seqlen=2048 | Seqlen=4096 |
> | :--- | :--- | :---: | :---: | :---: |
> | **Wider SwiGLU** | Speed (ms) | 81.8 | 171.0 | 338.4|
> | | Memory (MB) | 1991.5 | 3799.5 | 7415.5 |
> | **FlashMHF** | Speed (ms) | 94.2 | 180.2 | 347.6 |
> | | Memory (MB) | 619.8 | 1014.8 | 1847.0 |
>
> We observe that FlashMHF is slightly slower than the Wider SwiGLU baseline. However, we consider this an acceptable trade-off:
> 1.  **Baseline Issues:** The Wider SwiGLU suffers from a non-optimal expansion ratio ($d_{ff} \neq \frac{8}{3} \ d_{model}$) and reduced depth, likely degrading performance as discussed in Section 3.2.1.
> 2.  **Theoretical Advantage:** The speed gap is marginal compared to the significant Eval Loss gains. Theoretically, our I/O-aware kernel should be faster due to reduced memory traffic; the current lag is attributed to pending engineering optimizations since the SwiGLU Baseline is highly optimized on modern GPUs.
> 3. **Memory:** The memory gap remains massive. A wider SwiGLU requires materializing an even larger intermediate tensor ($L \times d_{ff}$), whereas FlashMHF processes this via tiling in SRAM.

---

> > ### Author Response · Authors · 2025-11-23
> >
> > ### **Response to Weakness 3 & Question 1: Memory Evaluation & Gradient Checkpointing**
> >
> > **The reviewer suggests that SwiGLU can be easily adapted to a flash kernel and questions whether our implementation accounts for optimizations like fusion and gradient checkpointing.**
> >
> > **1. Clarification on "Flash" vs. "Fusion"**
> > The reviewer suggests SwiGLU can be adapted to a flash kernel. We clarify that Fusion Kernel (e.g., xFormers, Liger Kernel) fuses element-wise ops (SiLU) but **cannot** fuse the two surrounding MatMuls into a single kernel without materializing the large intermediate activation to HBM. This is because the full intermediate dimension ($d_{ff}$) is too large to fit into the SRAM of a GPU Streaming Multiprocessor (SM).
> >
> > The following code snippet illustrates the Liger Kernel SwiGLU implementation. It primarily optimizes the SiLU operation by making it in-place. However, it does not prevent the materialization of the large intermediate tensor into HBM before the down projection, thus failing to achieve the memory and I/O advantages of a Flash-style kernel.
> >
> > ```python
> > class LigerSwiGLUMLP(nn.Module):
> >     def __init__(self, config):
> >         super().__init__()
> >         self.config = config
> >         self.hidden_size = config.hidden_size
> >         self.intermediate_size = config.intermediate_size
> >         self.gate_proj = nn.Linear(self.hidden_size, self.intermediate_size, bias=False)
> >         self.up_proj = nn.Linear(self.hidden_size, self.intermediate_size, bias=False)
> >         self.down_proj = nn.Linear(self.intermediate_size, self.hidden_size, bias=False)
> >         if config.hidden_act not in ["silu", "swish"]:
> >             raise ValueError(f"Activation function {config.hidden_act} not supported.")
> >
> >     def forward(self, x):
> >         return self.down_proj(LigerSiLUMulFunction.apply(self.gate_proj(x), self.up_proj(x)))
> > ```
> >
> > **2. Comparison of SwiGLU Variants vs. FlashMHF:**
> >
> > Testing on other optimized fused SwiGLU implementation (all under 4k sequence length), the results are as follows:
> >
> > | Metric | SwiGLU (Torch) | SwiGLU (XFormer) | SwiGLU (Liger Kernel) | FlashMHF |
> > | :--- | :---: | :---: | :---: | :---: |
> > | Speed(ms) | 365.32 | 516.81 | 333.17 | **329.72** |
> > | Memory(MB) | 6598.0 | 5180.0 | 5180.0 | **1682.2** |
> >
> > **Analysis:**
> > 1.  **Memory Efficiency:** The results demonstrate that traditional SwiGLU implementations, whether using standard PyTorch or fused kernels (xFormers or liger kernel), cannot reduce the peak memory usage of a single layer. In contrast, FlashMHF drastically reduces HBM write-back and memory I/O. By splitting the computation into heads, we tile the computation along the intermediate dimension, computing the `MatMul -> Activation -> MatMul` chain locally in SRAM **block by block**. This avoids materializing the full intermediate tensor to HBM, solving the memory bottleneck.
> > 2.  **Gradient Checkpointing:** We emphasize that Gradient Checkpointing is a technique primarily used during training to trade computation for memory. In contrast, our work prioritizes **inference**, which represents the dominant operational cost for modern LLMs. Training is fundamentally a one-time process, whereas inference occurs continuously at scale. FlashMHF directly addresses this critical inference bottleneck by drastically reducing memory usage and improving throughput. By solving the memory explosion issue, FlashMHF unlocks larger batch sizes and longer contexts, offering great value for large-scale deployment. Regarding training performance: On **pre-Hopper GPUs**, FlashMHF achieves faster training speed compared to the baseline, empirically validating our kernel design's theoretical speedup advantages. Our current implementation on **Hopper-architecture GPUs** (H100) is undergoing active kernel optimization to leverage Hopper-specific features. Importantly, even as we continue to refine training performance, this work establishes a promising research direction for bringing multi-head architectures to FFN layers with practical memory efficiency.

---

> > > ### Author Response · Authors · 2025-11-23
> > >
> > > ### **Response to Question 2: Necessity of Projection $W_{in}$**
> > >
> > > **In the multi-head FFN definition, a separate projection $W_{in}$ is introduced to obtain the query (Eq.10). Is this design choice empirically validated as necessary?**
> > >
> > > The linear projection prior to the head split ($W_{in}$ in Eq. 10) serves to mix information across the model dimension before separating it into independent heads. Similarly, the output projection ($W_{out}$ in Eq. 14) aggregates the processed information from these independent heads, mixing it back into the full model dimension ($d_{model}$). This ensures effective information exchange before and after the parallel processing stage.
> > >
> > > We further justify this design with two key insights. First, we previously did ablation studies, and the results shows that removing these projections while keeping the total parameter count constant, results in significant performance degradation, confirming their critical role in feature mixing. Second, this design mirrors the **Symmetry with Multi-Head Attention** mechanism, which uses $W_Q, W_K, W_V$ projections to mix features before splitting them into heads. Our approach maintains this structural symmetry, adhering to the standard "projection $\to$ split" paradigm.
> > >
> > >
> > > ### **Response to Question 3: Comparison with PAttention**
> > >
> > > **Compared to PKV, the activation function used in PAttention [1](https://arxiv.org/pdf/2410.23168) might serve as a more solid baseline for comparison.**
> > >
> > > We thank the reviewer for highlighting [*TokenFormer*](https://arxiv.org/pdf/2410.23168) and its PAttention mechanism, which is a very insightful work.
> > >
> > > We view our Parametric KV (PKV) baseline as the theoretical limit of our architecture where the sub-network dimension $d_e = 1$. The key is that PKV typically relies on Softmax-based activation, which introduces heavy competition across the hidden dimension. Our results indicate that the element-wise activation (SiLU) in FlashMHF avoids this bottleneck, maximizing parameter efficiency.
> > >
> > > However, a direct empirical comparison is challenging due to significant structural divergence. [TokenFormer](https://arxiv.org/pdf/2410.23168) modifies the entire Transformer block (replacing both Attention and MLP projections with complex non-linear transformations), whereas FlashMHF strictly modifies the FFN layer while preserving the standard Attention mechanism. Crucially, the PAttention study does not provide an ablation where *only* the FFN is replaced by PAttention, making it impossible to isolate the gains attributable to the FFN modification alone. Since *TokenFormer* alters the fundamental architecture extensively, we cannot perform a controlled variable comparison.
> > >
> > > Nevertheless, we acknowledge the shared conceptual "symmetry" in our approaches. We will cite this work and discuss this relationship in the revised Related Works.
> > >
> > > ### **Response to Question 4: Compatibility with MoE**
> > >
> > > **The reviewer notes that most SOTA Transformer architectures now adopt MoE designs and asks how the authors view the compatibility and potential integration of FlashMHF with MoE architectures.**
> > >
> > > FlashMHF is fully compatible with Mixture-of-Experts (MoE) architectures. Our parallel sub-network design is structurally equivalent to a **"Dense MoE"** (soft routing), which allows for seamless adaptation to a **"Sparse MoE"** (hard routing, Top-K) by simply modifying the router logic. We envision a future **"Shared Expert"** design where experts are pooled globally. For example, in a setup with 8 heads where each head originally accesses 8 distinct sub-networks, we can pool these to create a shared set of 64 experts. Each head can then sparsely select the top-8 experts from this global pool. This design significantly enhances parameter efficiency and model performance. While previous works like MH-MoE have shown the potential of Multi-Head MoEs, they suffered from memory explosion. FlashMHF complements this perfectly: our kernel solves the memory explosion issue, making scalable Multi-Head MoEs practical for large-scale deployment. We believe this architecture holds immense promise and have begun extensive experiments.

---

> ### Author Response · Authors · 2025-11-27
>
> Dear reviewer B4B2,
>
> We are grateful for your feedback and thank you again for the time and effort you put into reviewing our paper. We believe that our responses above fully address the points raised in the initial review. Since the discussion period is nearing its end, we kindly ask that you let us know whether you have any remaining questions or comments.
>
> Thank you!

---

### Official Review · Reviewer_i2pJ · 2025-11-01

**Soundness:** 3
**Presentation:** 2
**Contribution:** 3
**Rating:** 6
**Confidence:** 3

**Summary:**

This paper proposes FlashMHF, which is a multi-head feed-forward networks (FFNs) for Transformers. Motivated by the structural similarity between single-head attention and FFNs, the authors identify two key challenges in current MHF: memory explosion and scaling imbalance. FlashMHF solves these problems by pairing a scaled-balanced parallel FFN subnetworks designed with a high-efficiency, IO-aware kernel.  Experiments on models from 128M to 1.3B parameters show improvements in perplexity and downstream tasks, with 3-5x memory reduction and up to 1.08x inference speedup.

**Strengths:**

The motivation of the paper is well justified with two problems in naive multi-head attention. There are proper ablations such as head dimensions and model scales, and downstream task evaluations are standard. The idea is straightforward by using sub-networks to group different heads to solve the problems, yet results are pretty impressive.

**Weaknesses:**

1. In section 3.2.1 the authors say their FlashMHF functions Luke a dense MoE, however, there is no direct comparison against dense MoE architecture.
2. There is no ablations for “Flash”, so it’s hard to isolate memory savings from the architectural change and the kernel optimization.
3. Lack of large scale experiments to verify the scaling effect - largest model size is 1.3B.
4. About presentation, Figure 3a doesn’t show multihead which is confusing. Also, the biggest innovation of it seems to come from MoE, while the title is a bit misleading, “mixture of dense multi-head FFN experts” might be better cover what the core idea is.

**Questions:**

Multi-head needs to be concat so we do need to materialize the full tensor. In section 3.2.2 it says “The key to solving the memory explosion lies in the multi-head design itself” seems wrong, shouldn’t it be in the expert design, because we can do the weighted average accumulation?

---

> ### Author Response · Authors · 2025-11-23
>
> We thank the reviewer for the constructive feedback and the positive assessment of our work’s soundness and contribution. We have addressed the concerns regarding the comparison with Dense MoE, the mechanism of memory savings, and scalability below.
>
> ### **Response to Weakness 1: Comparison with "Dense MoE"**
>
> **The reviewer notes that in section 3.2.1 the authors say their FlashMHF functions like a dense MoE, however, there is no direct comparison against dense MoE architecture.**
>
> To address your concern, we conducted **new ablation experiments** on 128M and 370M scales where we use SwiGLU with Dense MoE (aka. parallel sub-networks) as FFN module (apply parallel sub-networks to SwiGLU without our specific multi-head structure).
>
> | Model Scale | Method | Eval Loss $\downarrow$ |
> | :--- | :--- | :--- |
> | 128M | SwiGLU Baseline | *3.286* |
> | 128M | SwiGLU with Dense MoE | 3.296 |
> | 128M | FlashMHF-128hdim | **3.258** |
> | 370M | SwiGLU Baseline | *3.030* |
> | 370M | SwiGLU with Dense MoE | 3.062 |
> | 370M | FlashMHF-128hdim | **3.014** |
>
> The SwiGLU with Dense MoE variant resulted in higher perplexity compared to the SwiGLU baseline, as well as FlashMHF. This validates our analysis on the aspect ratio. Splitting the SwiGLU reduces the effective $d_{ff}/d_{model}$ ratio far below the optimal range (approx. 8/3). FlashMHF’s parallel sub-network design is specifically engineered to restore this optimal ratio.
>
> **Conclusion:** The performance gains of FlashMHF stem from our specific architectural innovations (multi-head structure as well as scale-balanced sub-networks), not from applying a mixture-of-experts concept.

---

> ### Author Response · Authors · 2025-11-23
>
> ### **Response to Weakness 2 & Question: Source of Memory Savings**
>
> **The reviewer points out that there is no ablations for “Flash”, so it’s hard to isolate memory savings from the architectural change and the kernel optimization.**
>
> The memory saving comes directly from the **I/O-aware fused kernel optimization** (the "Flash" part). However, to support this kernel optimization on hardware, the **Multi-Head design is mandatory**. It is not derived from expert design.
>
> To illustrate this mechanism clearly, we provide a concrete calculation example below (assuming dimensions: $L=2, d_{model}=3, d_{ff}=6$, Tile size=2).
>
> **Definition:** Use ReLU FFN as example,
>
> $$
> \begin{aligned}
> O &= \text{ReLU}(Q \cdot K^T) \cdot V \\\\
> &= \text{ReLU}\left(
> \begin{bmatrix} 1 & 2 & 0 \\\\ 0 & 1 & 1 \end{bmatrix}
> \cdot
> \begin{bmatrix} 1 & 0 & 0 & -2 & 0 & 3 \\\\ 0 & 1 & 0 & 2 & -3 & 0 \\\\ 0 & 0 & 2 & 0 & -3 & -3 \end{bmatrix}
> \right)
> \cdot
> \begin{bmatrix} 1 & 0 & 0 \\\\ 0 & 1 & 0 \\\\ 0 & 0 & 2 \\\\ 2 & 2 & 0 \\\\ 0 & 3 & 3 \\\\ 3 & 0 & 3 \end{bmatrix}
> \\\\
> &= \begin{bmatrix} 14 & 6 & 9 \\\\ 4 & 5 & 4 \end{bmatrix}
> \end{aligned}
> $$
>
>
> **1) Naive Algorithm**
> First, compute $A = Q \cdot K^T$ and store it to HBM.
>
>
> $$
> A = Q \cdot K^T =
> \begin{bmatrix} 1 & 2 & 0 \\\\ 0 & 1 & 1 \end{bmatrix}
> \cdot
> \begin{bmatrix} 1 & 0 & 0 & -2 & 0 & 3 \\\\ 0 & 1 & 0 & 2 & -3 & 0 \\\\ 0 & 0 & 2 & 0 & -3 & -3 \end{bmatrix}
> = \\\\
> \begin{bmatrix} 1 & 2 & 0 & 2 & -6 & 3 \\\\ 0 & 1 & 2 & 2 & -6 & -3 \end{bmatrix}
> $$
>
>
> Then compute $\text{ReLU}(A) \cdot V$. The intermediate matrix $A \in L \times d_{ff}$ must be materialized in HBM. This creates high peak memory and lots of high-latency memory I/O.
>
> **2) Online Algorithm (I/O-aware online MLP)**
>
> Core Idea: We do not materialize the full matrix $A$. We process "one intermediate block at a time," accumulating the contribution directly to `partial O` in SRAM.
>
> **Step 1: Use Columns 1–2 of $K^T$ and Rows 1–2 of $V$**
> (Small enough to fit in SRAM, no write-back to HBM)
>
> $$
> \text{partial } O_1 = \text{ReLU}\left(
> \begin{bmatrix} 1 & 2 & 0 \\\\ 0 & 1 & 1 \end{bmatrix}
> \cdot
> \begin{bmatrix} 1 & 0 \\\\ 0 & 1 \\\\ 0 & 0 \end{bmatrix}
> \right)
> \cdot
> \begin{bmatrix} 1 & 0 & 0 \\\\ 0 & 1 & 0 \end{bmatrix}
> = \\\\
> \begin{bmatrix} 1 & 2 & 0 \\\\ 0 & 1 & 0 \end{bmatrix}
> $$
>
> All partial matrix (size $tile \times d{h}$) can be fit into SRAM.
>
> **Step 2: Use Columns 3–4 of $K^T$ and Rows 3–4 of $V$**
>
> $$
> \text{partial } O_2 = \text{ReLU}\left(
> \begin{bmatrix} 1 & 2 & 0 \\\\ 0 & 1 & 1 \end{bmatrix}
> \cdot
> \begin{bmatrix} -2 & 0 \\\\ 2 & -3 \\\\ 0 & -3 \end{bmatrix}
> \right)
> \cdot
> \begin{bmatrix} 2 & 2 & 0 \\\\ 0 & 3 & 3 \end{bmatrix}
> = \\\\
> \begin{bmatrix} 4 & 4 & 0 \\\\ 4 & 4 & 4 \end{bmatrix}
> $$
>
>
> **Step 3: Use Columns 5–6 of $K^T$ and Rows 5–6 of $V$**
>
> $$
> \text{partial } O_3 = \text{ReLU}\left(
> \begin{bmatrix} 1 & 2 & 0 \\\\ 0 & 1 & 1 \end{bmatrix}
> \cdot
> \begin{bmatrix} 0 & 3 \\\\ -3 & 0 \\\\ -3 & -3 \end{bmatrix}
> \right)
> \cdot
> \begin{bmatrix} 0 & 3 & 3 \\\\ 3 & 0 & 3 \end{bmatrix}
> = \\\\
> \begin{bmatrix} 9 & 0 & 9 \\\\ 0 & 0 & 0 \end{bmatrix}
> $$
>
> **Final Accumulation in SRAM:**
> The sum of these partial blocks yields the final $O$, during the computation, no write-back to HBM.
>
> $$
> O = \text{partial } O_1 + \text{partial } O_2 + \text{partial } O_3 =
> \begin{bmatrix} 1+4+9 & 2+4+0 & 0+0+9 \\\\ 0+4+0 & 1+4+0 & 0+4+0 \end{bmatrix}
> = \begin{bmatrix} 14 & 6 & 9 \\\\ 4 & 5 & 4 \end{bmatrix}
> $$
>
> Finally, `O` is written from SRAM to HBM. This saves the massive intermediate matrix write/read.
>
> **3) The Necessity of Multi-Head**
>
> In real-world large models, $d_{model}$ (e.g., 2048 or 4096) is too large. If we attempt to run the "Online Algorithm" on the full dimension, the partial O accumulator (size $L_{tile} \times d_{model}$) would exceed the limited on-chip SRAM capacity.
> By slicing the model into multiple heads (e.g., $d_{head}=128$), we ensure that the accumulator for each head ($L_{tile} \times 128$) fits entirely within SRAM. This allows the computation to remain "Flash" (IO-aware) without materializing the large $B \cdot L \cdot d_{ff}$ tensor to HBM.

---

> ### Author Response · Authors · 2025-11-23
>
> ### **Response to Weakness 3: Scalability**
>
> **The reviewer notes a lack of large scale experiments to verify the scaling effect - largest model size is 1.3B.**
>
> We appreciate the reviewer’s emphasis on scale. Our results (Original Table 1 and Table 2, and Table below) clearly demonstrate a **positive scaling trend**: the advantage of FlashMHF over the baseline *widens* as we scale from 128M to 370M, and is most significant at 1.3B. This empirical evidence suggests that FlashMHF's benefits are likely to increase, rather than diminish, at larger scales.
>
> | Model | 128M | 370M | 1.3B |
> | :--- | :---: | :---: | :---: |
> | SwiGLU | 3.286 | 3.030 | 2.843 |
> | FlashMHF | 3.258 | 3.014 | **2.793** |
> | Improvement| 0.028 | 0.016 | **0.050** |
>
> Regarding the experimental scale, we strictly followed the configurations of [Dao & Gu, 2024 ("Transformers are SSMs")](https://arxiv.org/pdf/2405.21060), establishing 1.3B trained on 100B tokens as a rigorous standard for validating novel architectures. As an academic laboratory, training a 1.3B model represents the upper limit of our available computational resources. However, the consistent and widening performance gap observed up to 1.3B provides strong evidence for the scalability of our approach.
>
>
> ### **Response to Weakness 4: Presentation**
>
> **The reviewer mentions that about presentation, Figure 3a doesn’t show multihead which is confusing.**
>
> We apologize for the confusion caused by the schematic. We have revised **Figure 3a** in the updated manuscript to explicitly visualize the multi-head parallel structure, ensuring the architecture is clear to readers.

---

> ### Author Response · Authors · 2025-11-27
>
> Dear reviewer i2pJ,
>
> We are grateful for your feedback and thank you again for the time and effort you put into reviewing our paper. We believe that our responses above fully address the points raised in the initial review. Since the discussion period is nearing its end, we kindly ask that you let us know whether you have any remaining questions or comments.
>
> Thank you!

---

### Author Response · Authors · 2025-12-03

Because our rebuttal was submitted relatively late, most reviewers did not have time to respond, and due to a subsequent system issue they are now unable to reply. To assist the Area Chair and Reviewers in the final evaluation, we summarize the status of the discussion and clarify the core contribution of our work.

# Summary of Contributions and Responses

## 1. Core Novelty
**a. Flash-Style Efficiency:** We treat the MLP intermediate dimension as sequence length. By introducing a multi-head mechanism, we fit the computation into SRAM, enabling online processing without materializing large activations to HBM.

**b. Solving Scaling Imbalance:** To fix the imbalanced $d_{ff}/d_{head}$ ratio in naive multi-head MLPs, we use a sub-network routing mechanism (like MoE). This restores the optimal aspect ratio for robust scaling.

**c. Hardware-Software Co-Design:**
Through this hardware-software co-design, we achieve superior performance (lower perplexity) while significantly reducing memory usage compared to standard baselines.

---

## 2. Key Concerns

### 1. Comparison with Dense MoE
We added a new ablation experiment, which shows that FlashMHF outperforms the "Dense MoE" baseline (SwiGLU + sub-networks, no multi-head), proving our multi-head architecture is the source of gains.

| Model Scale | Method | Eval Loss $\downarrow$ |
| :--- | :--- | :--- |
| **128M** | SwiGLU Baseline | 3.286 |
| | SwiGLU with Dense MoE | 3.296 |
| | **FlashMHF** | **3.258** |
| **370M** | SwiGLU Baseline | 3.030 |
| | SwiGLU with Dense MoE | 3.062 |
| | **FlashMHF** | **3.014** |

### 2. Memory Advantage
We tile computation along the intermediate dimension (like FlashAttention), executing the full `MatMul -> Activation -> MatMul` chain in SRAM. This avoids expensive HBM I/O for the large intermediate tensor.

### 3. Comparison with Gradient Checkpointing (GC)
Gradient Checkpointing (GC) is ~1.5x slower due to re-computation, whereas FlashMHF offers "free" memory savings with no overhead on non-Hopper architectures. Additionally, FlashMHF improves perplexity, while GC is merely a system optimization. Our training speed matches optimized fused MLPs on non-Hopper GPUs, validating our theoretical approach; optimization for Hopper is a pending engineering task.

---

### Meta-Review · Area_Chair_ShwU · 2025-12-28

**Summary:**

The main concerns are around comparison with dense MoEs, experiments at large scale, memory evaluation (in particular comparison to gradient checkpointing), and limited novelty. The authors showed experiments for memory evaluation and justified the statements regarding subnetwork improvements and speed evaluation. However, the novelty is still limited, and the scaling part beyond 1.3B is still largely.

**Reviewer Concerns:**

Addressed:

1. Memory evaluation and comparison to gradient checkpointing are shown and discussed in the rebuttal.
2. Improvement when model size grows (up to 1.3B)
3. Comparison with dense MoE.

Still outstanding:
1. justification of novelty.
2. larger models beyond 1.3B.

**Reviewer Scores:**

Reviewers might keep their scores after rebuttal.

---

### Decision · Program_Chairs · 2026-01-26

Reject